# THE POWER OF MINIMALISM IN LONG-TERM TIME SERIES FORECASTING

## ABSTRACT

Recently, transformer-based models have been widely applied to time series fore-casting tasks due to their remarkable capability to capture complex interactions within sequential data. However, as the sequence length expands, Transformer-based models suffer from increased memory consumption, overfitting, and perfor-mance deterioration in capturing long-range dependencies. Recently, several stud-ies have shown that MLP-based models can outperform advanced Transformer-based models for long-term time series forecasting (LTSF) tasks. Unfortunately, linear mappings often struggle to capture intricate dependencies when handling multivariate time series. Although modeling each channel independently can al-leviate this issue, it will significantly increase the computational cost. To this end, we introduce a set of simple yet effective depthwise convolution models named `LTSF-Conv` to perform LTSF. Specifically, we apply unique filters to each chan-nel to achieve channel independence, which plays a pivotal role in enhancing over-all forecasting performance. Experimental results show that `LTSF-Conv` models outperform the state-of-the-art Transformer-based and MLP-based models across seven real-world LTSF benchmarks. Surprisingly, a two-layer non-stacked net-work can outperform the state-of-the-art Transformer model in 91% of cases with a significant reduction of computing resources. In particular, `LTSF-Conv` mod-els substantially decrease the average number of trainable parameters (by $\sim 12\times$), maximum memory consumption (by $\sim 86\times$), running time (by $\sim 18\times$), and infer-ence time (by $\sim 2\times$) on the Electricity benchmark. We hope this simple network unit opens up new research directions for the LTSF tasks.

## 1 INTRODUCTION

Time series forecasting is a prevalent problem across a series of domains, including energy man-agement for efficient resource allocation (Shao et al., 2022b), meteorology for weather forecasting (Wang et al., 2019), transportation for optimizing traffic flow (Shao et al., 2022a), and finance for in-formed decision-making (Cheng et al., 2022). These applications emphasize the far-reaching impact of accurate predictions in the real world. In the era of data-driven decision-making, extending the forecasting horizon can enable organizations and policymakers to plan long-term goals and strate-gies more accurately. In this context, long-term time series forecasting (LTSF) emerges as an area of paramount significance. However, extending the length of time series input-output presents several challenges in modeling techniques for LTSF tasks, triggering significant interest among researchers (Zhou et al., 2021; Wu et al., 2021; Zhou et al., 2022; Zeng et al., 2022).

In recent years, the rapid development of deep learning technology has led to significant progress in time series forecasting tasks. Transformer-based models have attracted considerable attention among various approaches. The self-attention mechanism in Transformers enables it to effectively capture long-term dependencies and dynamically focus on different parts of the input sequence. Although Transformers have demonstrated exceptional performance in tasks like natural language processing (NLP) (Devlin et al., 2018; Brown et al., 2020), transferring their success to the time series domain presents challenges. The permutation-invariant self-attention mechanism in Trans-formers, although powerful, can result in some loss of temporal information. Some improved posi-tional embeddings are added to convey temporal information, but they might not fully compensate for the inherent sequence ordering present in time series data, inadvertently leading to a loss of fine-grained temporal information that is vital for accurate forecasting. Moreover, the substantial

size of Transformer models leads to longer training times and increased computational demands, especially during the training process where the risk of overfitting becomes more high. This hypothesis has received empirical validation in a study referenced as (Zeng et al., 2022), wherein a very simple linear model, named DLinear, managed to outperform the majority of the Transformer-based forecasting models mentioned earlier. This emphasizes the significance of considering model complexity, as even a seemingly basic model can yield competitive results in comparison to more intricate architectures. Built on this research, a series of subsequent approaches have arisen, focusing on the utilization of Multi-Layer Perceptron (MLP) based techniques. Although these stacked linear layer approaches slightly improve the predictive performance, MLP-based models often possess constrained representational capacity, especially when dealing with multi-channel datasets. A potential solution involves the adoption of channel-independent (CI) modeling, as proposed in reference (Nie et al., 2023). However, it concurrently introduces a substantial increase in computational demands. Among various deep learning models developed for long-term time series forecasting, it's difficult to strike a balance between performance and efficiency when using Transformer-based models or MLP-based models.

Convolutional neural networks possess an underestimated potential in the domain of long-term time series prediction, providing a promising path for addressing current issues. To answer this question, we engage in an investigation involving intensive experiments and analyses to delve into the intricate operational mechanisms of recent LTSF models. We then introduce a series of models named `LTSF-Conv`, which integrate an exceedingly simple basic network. Our models utilize a single depthwise convolution layer as the temporal feature extractor and employ a one-layer linear projection for the final prediction results. We perform comprehensive experiments on seven public real-world datasets across several domains: weather, traffic, electricity, etc. Surprisingly, our results reveal that `LTSF-Conv` models consistently outperform existing complex Transformer-based models for averaged performance across all scenarios in 91% cases. Furthermore, our models also surpass the state-of-the-art stacked MLP-based models for averaged performance in 86% cases. We find that, in contrast to the existing transformer-based and MLP-based architectures, `LTSF-Conv` models exhibit remarkable performance while maintaining a high level of efficiency. The main contributions of this work include:

- We have introduced a series of two-layer convolution models, called `LTSF-Conv`. It's important to emphasize that these models maintain an unstacked structure, representing a simple yet competitive basic unit with substantial potential for future research exploration in LTSF tasks.

- Surprisingly, experimental evaluations on seven popular public datasets show that our simple foundational component significantly outperforms the strongest Transformer-based and MLP-based models in LTSF tasks while achieving notable reductions in compute resources. For example, `Conv` model attains the best results in 100% of cases for average metrics across various horizons, while achieving a 95% reduction in the number of training parameters, a 99% reduction in GPU memory usage, a 90% reduction in running time, and a 54% reduction in inference time compared to the strongest Transformer model on the Weather benchmark.

- We conduct comprehensive empirical investigations across some dimensions of popular Transformer-based and MLP-based solutions, including the sensitivity to different input window sizes, the impact of encoder-decoder structure, and the limitations in dealing with time series with multiple periods among channels, etc. These findings can offer valuable insights for future research within the realm of LTSF.

## 2 BACKGROUNDS

### 2.1 TRANSFORMER-BASED LTSF MODELS

Over the past few years, researchers have dedicated significant effort to propose novel Transformer models for improving long sequence prediction performance. However, dealing with long sequences can drastically increase the computational complexity of the Transformer-based models, resulting in a greater demand for computational resources and processing time. To address the challenge from the quadratic complexity inherent in self-attention mechanisms, the LogTrans (Li et al., 2019)

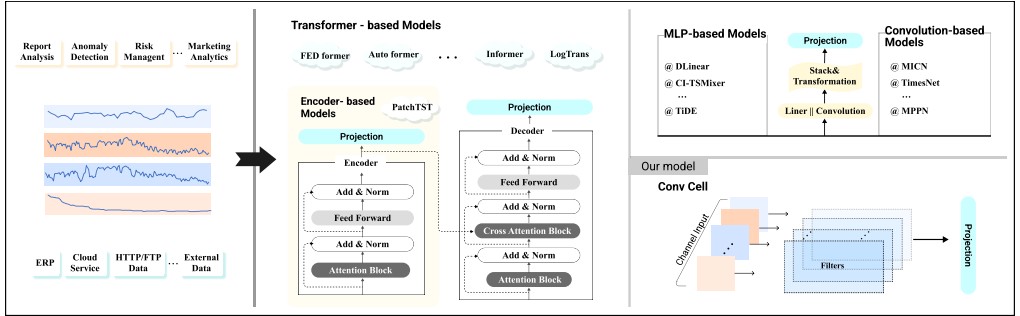

Figure 1: The pipeline of existing LTSF solutions.

model introduces an innovative attention mechanism called LogSparse attention, which effectively mitigates computational complexity, reducing it to $\mathcal{O}(L(logL)^2)$. Similarly, the Informer model (Zhou et al., 2021) incorporates the distilling operation and the ProbSparse attention mechanism based on the Kullback-Leibler divergence. These strategies further significantly reduce the computational complexity to $\mathcal{O}(L(logL))$. Autoformer (Wu et al., 2021) introduces an auto-correlation mechanism to enhance computational efficiency and the utilization of sequence information. Furthermore, it also incorporates a decomposition block that can progressively extract the long-term stationary trend. Building upon this scheme, FEDFormer (Zhou et al., 2022) improves decomposed blocks by integrating frequency domain representations of time series into attention computation. The forecasting accuracy is significantly influenced by the size of the look-back window. To capture more contextual information from a longer historical window, the PatchTST model (Nie et al., 2023) presents a patching component that segments the time series into subseries-level patches. PatchTST first applies channel-independence technology to Transformer-based models and significantly improves prediction accuracy.

## 2.2 Convolution and MLP-based LTSF models

Although Transformer-based models exhibit potential in time series forecasting, challenges arise from the disparities between NLP and time series data. Recently, DLinear (Zeng et al., 2022) achieved remarkable success by relying on a single linear layer, surpassing the performance of certain complex Transformer-based architectures by a significant margin. On top of this scheme, subsequent MLP-based approaches exhibit better performance in LTSF tasks. TSMixer (Vijay et al., 2023) presents a hybrid architecture of augmenting various reconciliation heads that enhances the learning capacity by integrating gated attention mechanisms into the channel-independent backbone. TiDE (Das et al., 2023) is a simple encoder-decoder model that encodes historical time-series data and covariates using dense MLPs, followed by decoding the time-series and future covariates using dense MLPs again. Many of the extracted patterns might encompass unpredictable noise and lack clear interpretability. Besides MLP-based solutions, other types of convolution networks are also widely explored. SCINet (Liu et al., 2022) employs a recursive downsample-convolve-interact architecture that iteratively downsamples the sequence into two sub-sequences to effectively extract temporal features. (Wu et al., 2022) proposes a task-general foundational model named TimesNet, which transforms 1D time series into a set of 2D tensors to more effectively capture intra-period and inter-period variations in the 2D space. MICN (Shao et al., 2022b) captures local correlations and global correlations by incorporating a multi-scale branch structure with down-sampled convolution. MPPN (Wang et al., 2023) constructs multi-resolution contextual-aware semantic units within time series and introduces a multi-period pattern mining mechanism to explicitly capture crucial time series patterns, which has better interpretability.

## 3 Theoretical Study on the Convolution Mapping

### 3.1 Notation

Considering a series of observed time series signals $X = [x_0, \ldots, x_t, \ldots, x_{L-1}] \in \mathbb{R}^{C \times L}$, where $L$ represents the length of time steps and $C$ denotes the recorded variates. Then, given a historical time

series $X_{in} = [x_{t-sl}, \ldots, x_{t-1}]$, our goal is to learn a mapping function and forecast the subsequent $T$ time steps $\hat{Y} = [x_t, \ldots, x_{t+T-1}] \in \mathbb{R}^{C \times T}$.

## 3.2 THEORETICAL INSIGHTS

In this paper, we consistently employ the following variable names: $sl \leq L$: the look-back window length, $kl \leq sl$: the convolutional kernel length, $kn$: the number of convolution kernels. The definition of a convolution operation is as follows

$$Z = K \otimes X_{in}, \tag{1}$$

where $Z \in \mathbb{R}^{C \times sl}$. The $h$-$th$ element of the output sequence can be expressed as follows

$$z[h] = \sum_{i=0}^{kl-1} (w(i) * x(h+i)) + b, \tag{2}$$

where $w$ represents the weights associated with the convolutional window, $i$ is the relative position of the convolution window within the input sequence, and $b$ is the bias term.

For time series forecasting tasks, periodicity and trend with tolerable noise are fundamental characteristics that play a crucial role in the success of time series forecasting (Holt, 2004) (Zhang & Qi, 2005). Periodicity refers to the data exhibiting regular and predictable fluctuations over a specific interval. By capturing the periodic behavior, forecasting models can identify repetitive patterns. The trend represents the underlying upward or downward movement observed in the series. Incorporating the trend component is essential in forecasting as it helps capture the overall trajectory and anticipate future changes in the data. We first consider the widely acknowledged assumption that the time series exhibits periodicity.

**Theorem 1.** *Given a seasonal time series that satisfies $x(t) = x(t-p)$, where $p \leq kl$ is the period. A solution exists for valid convolution models to predict future values, which can be expressed as follows*

$$w(i) = \begin{cases} 1, & \text{if } i = (sl - \alpha \cdot p) \bmod p \\ 0, & \text{otherwise} \end{cases}, b_i = 0, \tag{3}$$

where $1 \leq \alpha \in \mathbb{Z} \leq \lfloor sl/p \rfloor$. Equation 3 implies that when the length of the input historical sequence and the convolutional kernel are equal to or greater than the period, convolutional maps have the ability to predict periodic signals. More proof is in Appendix A.

## 4 CONVOLUTION-BASED LTSF SOLUTIONS

`LTSF-Conv` is a set of convolution-based models. For clearness, we name these two types of temporal variations as `Conv` and `DConv` respectively.

### 4.1 BASIC BACKBONE

We denote the $i$-$th$ univariate series with a length of $L$, starting at time index 0, as $x_{0:L-1}^i = (x_0^i, x_1^i, \ldots, x_{L-1}^i)$, where $i$ ranges from 1 to $C$. The input is divided into $C$ separate univariate series, where each series is independently fed into a separate kernel backbone. In LTSF tasks, maintaining channel independence has been observed to improve prediction performance as compared to channel mixing. Concretely, the input time series firstly segment undergoes reversible instance normalization, denoted as RevIN (Nie et al., 2023). RevIN is applied to standardize the data distribution by removing the mean and dividing by the standard deviation, a process designed to mitigate data shifts within the time series.

`Conv` uses a one-layer depthwise convolution operation, and each convolutional kernel is responsible for processing one input channel independently. Each channel is convolved with a separate kernel, and this process generates a feature map with the same number of channels as the input. We utilizes group-wise convolution to cleverly achieve channel independence. Then, we apply a one-layer linear projection to the feature map for the final prediction results.

Autoformer (Wu et al., 2021) first introduces a new methodology by integrating seasonal-trend decomposition ahead of each neural block, leveraging a widely employed technique in time series analysis (Cleveland et al., 1990; Hamilton, 2020) for enhancing the predictability of raw data. Inspired by this scheme, `DConv` combines a decomposition scheme with convolutional layers to create a powerful and lightweight model architecture. `DConv` initiates the modeling process by decomposing the raw input data into two main components: the trend component and the seasonal component. Then, we use the separate one-layer linear projection to each component for the final prediction results, as follows

$$X_t = Z^\mathsf{T}, \tag{4}$$

$$X_s = X - X_t, \tag{5}$$

$$\hat{Y} = (W_t(X_t) + W_s(X_s))^\mathsf{T}, \tag{6}$$

where $Z$ can be obtained by Equation 1, and $W_t$ and $W_s \in \mathbb{R}^{T \times sl}$ represent the one-layer linear model. This process allows the model to model the unique characteristics and patterns within each component individually. Subsequently, the features obtained from both components are summed up to obtain the final prediction. By incorporating these two linear layers and aggregating their outputs, `DConv` effectively combines the information from both the trend and remainder components to make a comprehensive prediction.

### 4.1.1 Loss Function

We choose to use the mean squared error (MSE) loss as a measure of the discrepancy between the predictions and the ground truth in each channel. The MSE loss in each channel, denoted as $||\hat{y}_{sl:sl+T-1}^i - y_{sl:sl+T-1}^i||_2^2$, is computed individually for each time series. Then, the losses from $C$ time series are gathered and averaged to obtain the overall objective loss, as follows

$$\mathcal{L} = \mathbb{E}_X \frac{1}{C} \sum_{i=1}^{C} ||\hat{y}_{sl:sl+T-1}^i - y_{sl:sl+T-1}^i||_2^2 \tag{7}$$

## 5 Experimental Evaluation

### 5.1 Dataset

In this section, we assess the performance of the `LTSF-Conv` models on seven widely used multivariate time series datasets, including Weather, Traffic, Electricity, and four ETT datasets. Table 6 of the Appendix provides a brief overview of these datasets. For the Weather, Electricity, and Traffic datasets, adopted a standard split ratio of 7:1:2 for the training set, validation set, and test set, respectively. For the ETT dataset, adopted a standard split ratio of 6:2:2 for the training set, validation set, and test set. Since our primary interest lies in long-term forecasting results, we have excluded the ILI dataset with shorter horizons.

### 5.2 Baselines and Configuration

We choose recent popular Transformer models in the domain of LTSF for comparison, including PatchTST (Nie et al., 2023), Preformer (Du et al., 2023), FEDformer (Zhou et al., 2022), and Autoformer (Wu et al., 2021). Additionally, we also compare the latest non-Transformer models: MPPN (Wang et al., 2023), TimesNet (Wu et al., 2022), MICN-regre (Shao et al., 2022b), TiDE (Das et al., 2023), and DLinear (Zeng et al., 2022). By default, all methods adhere to a uniform experimental setup with prediction lengths of 96, 192, 336, 720. We utilize Mean Squared Error (MSE) and Mean Absolute Error (MAE) as the standard error metrics for evaluation. To ensure fairness in comparison, in Table 1 and Table 4, we perform experiments with a fixed look-back window size of 512.

Table 1: Multivariate long-term time series forecasting results in terms of MSE and MAE, the lower the better. The best numbers in each row are highlighted in bold and the second best numbers are highlighted with an underline. Among them, * denotes re-implementation with a fixed look-back window of 512. Other results collect from PatchTST (Nie et al., 2023).

| Methods | | Conv* (Ours) | | DConv* (Ours) | | PatchTST 2023 | | TimesNet* 2022 | | MICN-regre* 2022 | | FEDformer 2022 | | Autoformer 2022 | | DLinear 2022 | | Informer 2021 | |
|---|---|---|---|---|---|---|---|---|---|---|---|---|---|---|---|---|---|---|---|---|
| Metric | | MSE | MAE | MSE | MAE | MSE | MAE | MSE | MAE | MSE | MAE | MSE | MAE | MSE | MAE | MSE | MAE | MSE | MAE |
| Weather | 96 | 0.140 | 0.190 | 0.169 | 0.223 | 0.149 | 0.198 | 0.159 | 0.215 | 0.174 | 0.236 | 0.238 | 0.314 | 0.249 | 0.329 | 0.176 | 0.237 | 0.354 | 0.405 |
| | 192 | 0.183 | 0.230 | 0.214 | 0.260 | 0.194 | 0.241 | 0.220 | 0.267 | 0.221 | 0.282 | 0.275 | 0.329 | 0.325 | 0.37 | 0.220 | 0.282 | 0.419 | 0.434 |
| | 336 | 0.234 | 0.271 | 0.259 | 0.294 | 0.245 | 0.282 | 0.274 | 0.306 | 0.257 | 0.316 | 0.339 | 0.377 | 0.351 | 0.391 | 0.265 | 0.319 | 0.583 | 0.543 |
| | 720 | 0.306 | 0.325 | 0.324 | 0.342 | 0.314 | 0.334 | 0.347 | 0.356 | 0.319 | 0.359 | 0.389 | 0.409 | 0.415 | 0.426 | 0.323 | 0.362 | 0.916 | 0.705 |
| | Avg | 0.216 | 0.254 | 0.242 | 0.280 | 0.225 | 0.263 | 0.250 | 0.286 | 0.243 | 0.298 | 0.310 | 0.357 | 0.335 | 0.379 | 0.246 | 0.300 | 0.568 | 0.521 |
| Electricity | 96 | 0.132 | 0.227 | 0.135 | 0.231 | 0.129 | 0.222 | 0.181 | 0.286 | 0.155 | 0.265 | 0.186 | 0.302 | 0.196 | 0.313 | 0.140 | 0.237 | 0.304 | 0.393 |
| | 192 | 0.145 | 0.241 | 0.146 | 0.243 | 0.147 | 0.240 | 0.192 | 0.294 | 0.185 | 0.293 | 0.197 | 0.311 | 0.211 | 0.324 | 0.153 | 0.249 | 0.327 | 0.417 |
| | 336 | 0.161 | 0.257 | 0.163 | 0.259 | 0.163 | 0.259 | 0.196 | 0.299 | 0.180 | 0.292 | 0.213 | 0.328 | 0.214 | 0.327 | 0.169 | 0.267 | 0.333 | 0.422 |
| | 720 | 0.201 | 0.289 | 0.203 | 0.292 | 0.197 | 0.290 | 0.226 | 0.323 | 0.207 | 0.316 | 0.233 | 0.344 | 0.236 | 0.342 | 0.203 | 0.301 | 0.351 | 0.427 |
| | Avg | 0.160 | 0.254 | 0.162 | 0.256 | 0.159 | 0.252 | 0.199 | 0.301 | 0.182 | 0.292 | 0.207 | 0.321 | 0.214 | 0.326 | 0.166 | 0.263 | 0.328 | 0.414 |
| ETTm2 | 96 | 0.161 | 0.249 | 0.163 | 0.252 | 0.166 | 0.256 | 0.191 | 0.275 | 0.176 | 0.271 | 0.180 | 0.271 | 0.205 | 0.293 | 0.167 | 0.260 | 0.355 | 0.462 |
| | 192 | 0.216 | 0.288 | 0.218 | 0.289 | 0.223 | 0.296 | 0.257 | 0.324 | 0.254 | 0.334 | 0.252 | 0.318 | 0.278 | 0.336 | 0.224 | 0.303 | 0.595 | 0.586 |
| | 336 | 0.271 | 0.327 | 0.271 | 0.325 | 0.274 | 0.329 | 0.295 | 0.349 | 0.288 | 0.351 | 0.324 | 0.364 | 0.343 | 0.379 | 0.281 | 0.342 | 1.270 | 0.871 |
| | 720 | 0.361 | 0.387 | 0.357 | 0.380 | 0.362 | 0.385 | 0.452 | 0.432 | 0.417 | 0.440 | 0.410 | 0.420 | 0.414 | 0.419 | 0.397 | 0.421 | 3.001 | 1.267 |
| | Avg | 0.252 | 0.313 | 0.252 | 0.312 | 0.256 | 0.316 | 0.299 | 0.346 | 0.284 | 0.350 | 0.291 | 0.343 | 0.310 | 0.356 | 0.267 | 0.331 | 1.305 | 0.796 |
| ETTm1 | 96 | 0.292 | 0.338 | 0.304 | 0.346 | 0.293 | 0.346 | 0.335 | 0.376 | 0.311 | 0.364 | 0.326 | 0.390 | 0.510 | 0.492 | 0.299 | 0.343 | 0.626 | 0.560 |
| | 192 | 0.332 | 0.361 | 0.335 | 0.365 | 0.333 | 0.370 | 0.396 | 0.410 | 0.356 | 0.388 | 0.365 | 0.415 | 0.514 | 0.495 | 0.335 | 0.365 | 0.725 | 0.619 |
| | 336 | 0.364 | 0.380 | 0.366 | 0.383 | 0.369 | 0.392 | 0.441 | 0.433 | 0.407 | 0.422 | 0.392 | 0.425 | 0.510 | 0.492 | 0.369 | 0.386 | 1.005 | 0.741 |
| | 720 | 0.418 | 0.411 | 0.419 | 0.412 | 0.416 | 0.420 | 0.551 | 0.495 | 0.464 | 0.462 | 0.446 | 0.458 | 0.527 | 0.493 | 0.425 | 0.421 | 1.133 | 0.845 |
| | Avg | 0.352 | 0.373 | 0.356 | 0.377 | 0.352 | 0.382 | 0.431 | 0.429 | 0.408 | 0.425 | 0.382 | 0.422 | 0.515 | 0.493 | 0.357 | 0.378 | 0.872 | 0.691 |
| ETTh1 | 96 | 0.365 | 0.393 | 0.366 | 0.392 | 0.37 | 0.400 | 0.443 | 0.458 | 0.398 | 0.424 | 0.376 | 0.415 | 0.435 | 0.446 | 0.375 | 0.399 | 0.941 | 0.769 |
| | 192 | 0.401 | 0.416 | 0.400 | 0.412 | 0.413 | 0.429 | 0.492 | 0.491 | 0.481 | 0.495 | 0.423 | 0.446 | 0.456 | 0.457 | 0.405 | 0.416 | 1.007 | 0.786 |
| | 336 | 0.419 | 0.437 | 0.420 | 0.424 | 0.422 | 0.440 | 0.483 | 0.487 | 0.516 | 0.524 | 0.444 | 0.462 | 0.486 | 0.487 | 0.439 | 0.443 | 1.038 | 0.784 |
| | 720 | 0.464 | 0.472 | 0.435 | 0.451 | 0.447 | 0.468 | 0.545 | 0.520 | 0.743 | 0.664 | 0.469 | 0.492 | 0.515 | 0.517 | 0.472 | 0.490 | 1.144 | 0.857 |
| | Avg | 0.412 | 0.430 | 0.405 | 0.420 | 0.413 | 0.434 | 0.491 | 0.489 | 0.532 | 0.527 | 0.428 | 0.453 | 0.473 | 0.476 | 0.422 | 0.437 | 1.032 | 0.799 |
| ETTh2 | 96 | 0.269 | 0.339 | 0.269 | 0.334 | 0.274 | 0.337 | 0.383 | 0.420 | 0.313 | 0.396 | 0.332 | 0.374 | 0.332 | 0.368 | 0.289 | 0.353 | 1.549 | 0.952 |
| | 192 | 0.329 | 0.383 | 0.330 | 0.376 | 0.341 | 0.382 | 0.405 | 0.437 | 0.498 | 0.519 | 0.407 | 0.446 | 0.426 | 0.434 | 0.383 | 0.418 | 3.792 | 1.542 |
| | 336 | 0.335 | 0.394 | 0.327 | 0.387 | 0.329 | 0.384 | 0.390 | 0.437 | 0.929 | 0.705 | 0.400 | 0.447 | 0.477 | 0.479 | 0.448 | 0.465 | 4.215 | 1.642 |
| | 720 | 0.379 | 0.424 | 0.381 | 0.432 | 0.379 | 0.422 | 0.460 | 0.471 | 1.256 | 0.817 | 0.412 | 0.469 | 0.453 | 0.490 | 0.605 | 0.551 | 3.656 | 1.619 |
| | Avg | 0.328 | 0.385 | 0.327 | 0.382 | 0.330 | 0.381 | 0.410 | 0.441 | 0.749 | 0.609 | 0.387 | 0.434 | 0.422 | 0.442 | 0.431 | 0.446 | 3.303 | 1.438 |
| Traffic | 96 | 0.396 | 0.275 | 0.391 | 0.272 | 0.360 | 0.249 | 0.602 | 0.321 | 0.473 | 0.306 | 0.576 | 0.359 | 0.597 | 0.371 | 0.410 | 0.282 | 0.733 | 0.410 |
| | 192 | 0.407 | 0.279 | 0.396 | 0.273 | 0.379 | 0.256 | 0.615 | 0.331 | 0.475 | 0.298 | 0.610 | 0.380 | 0.607 | 0.382 | 0.423 | 0.287 | 0.777 | 0.435 |
| | 336 | 0.417 | 0.285 | 0.408 | 0.281 | 0.392 | 0.264 | 0.614 | 0.333 | 0.493 | 0.307 | 0.608 | 0.375 | 0.623 | 0.387 | 0.436 | 0.296 | 0.776 | 0.434 |
| | 720 | 0.453 | 0.304 | 0.451 | 0.301 | 0.432 | 0.286 | 0.648 | 0.348 | 0.531 | 0.325 | 0.621 | 0.375 | 0.639 | 0.395 | 0.466 | 0.315 | 0.827 | 0.466 |
| | Avg | 0.426 | 0.286 | 0.412 | 0.282 | 0.390 | 0.263 | 0.620 | 0.333 | 0.493 | 0.309 | 0.603 | 0.372 | 0.616 | 0.383 | 0.433 | 0.295 | 0.778 | 0.436 |

Table 2: Multivariate long-term time series forecasting results. *Best* denotes re-implementation with a longer look-back window of {336, 512, 720, 1600}, consistently choosing the most optimal results. Other results collect from Tide (Nie et al., 2023) and MPPN (Wang et al., 2023).

| Methods | Metric | Weather | | | | | Electricity | | | | | ETTm2 | | | | | ETTm1 | | | | |
|---|---|---|---|---|---|---|---|---|---|---|---|---|---|---|---|---|---|---|---|---|---|
| | | 96 | 192 | 336 | 720 | Avg | 96 | 192 | 336 | 720 | Avg | 96 | 192 | 336 | 720 | Avg | 96 | 192 | 336 | 720 | Avg |
| Conv-Best | MSE | 0.140 | 0.182 | 0.234 | 0.294 | 0.213 | 0.129 | 0.143 | 0.159 | 0.195 | 0.157 | 0.161 | 0.217 | 0.257 | 0.325 | 0.240 | 0.287 | 0.328 | 0.364 | 0.394 | 0.343 |
| | MAE | 0.188 | 0.230 | 0.271 | 0.324 | 0.253 | 0.225 | 0.238 | 0.255 | 0.286 | 0.251 | 0.249 | 0.287 | 0.329 | 0.379 | 0.311 | 0.334 | 0.358 | 0.380 | 0.408 | 0.370 |
| Dconv-Best | MSE | 0.166 | 0.209 | 0.253 | 0.306 | 0.234 | 0.130 | 0.144 | 0.160 | 0.199 | 0.158 | 0.161 | 0.213 | 0.258 | 0.325 | 0.239 | 0.300 | 0.335 | 0.356 | 0.393 | 0.346 |
| | MAE | 0.220 | 0.259 | 0.293 | 0.335 | 0.277 | 0.225 | 0.238 | 0.256 | 0.288 | 0.252 | 0.253 | 0.292 | 0.325 | 0.369 | 0.310 | 0.342 | 0.363 | 0.384 | 0.405 | 0.374 |
| PatchTST-Best | MSE | 0.149 | 0.194 | 0.245 | 0.306 | 0.224 | 0.129 | 0.147 | 0.163 | 0.197 | 0.159 | 0.165 | 0.220 | 0.274 | 0.362 | 0.252 | 0.290 | 0.332 | 0.366 | 0.407 | 0.349 |
| | MAE | 0.198 | 0.241 | 0.282 | 0.334 | 0.263 | 0.222 | 0.240 | 0.259 | 0.290 | 0.252 | 0.255 | 0.292 | 0.329 | 0.380 | 0.314 | 0.342 | 0.369 | 0.392 | 0.421 | 0.381 |
| Preformer-Best | MSE | 0.227 | 0.275 | 0.324 | 0.394 | 0.305 | 0.180 | 0.189 | 0.201 | 0.232 | 0.201 | 0.213 | 0.269 | 0.324 | 0.418 | 0.306 | 0.516 | 0.556 | 0.572 | 0.598 | 0.561 |
| | MAE | 0.292 | 0.322 | 0.352 | 0.393 | 0.340 | 0.297 | 0.302 | 0.319 | 0.342 | 0.315 | 0.295 | 0.329 | 0.363 | 0.416 | 0.351 | 0.482 | 0.491 | 0.509 | 0.533 | 0.504 |
| TiDE | MSE | 0.166 | 0.209 | 0.254 | 0.313 | 0.235 | 0.132 | 0.147 | 0.161 | 0.196 | 0.159 | 0.161 | 0.215 | 0.267 | 0.352 | 0.248 | 0.306 | 0.335 | 0.364 | 0.413 | 0.354 |
| | MAE | 0.222 | 0.263 | 0.301 | 0.340 | 0.281 | 0.229 | 0.243 | 0.261 | 0.294 | 0.256 | 0.251 | 0.289 | 0.326 | 0.383 | 0.312 | 0.349 | 0.366 | 0.384 | 0.413 | 0.378 |
| MPPN | MSE | 0.144 | 0.189 | 0.240 | 0.310 | 0.220 | 0.131 | 0.145 | 0.162 | 0.200 | 0.159 | 0.162 | 0.217 | 0.273 | 0.368 | 0.255 | 0.287 | 0.330 | 0.369 | 0.426 | 0.353 |
| | MAE | 0.196 | 0.240 | 0.281 | 0.333 | 0.262 | 0.226 | 0.239 | 0.256 | 0.289 | 0.252 | 0.250 | 0.288 | 0.325 | 0.383 | 0.311 | 0.335 | 0.360 | 0.382 | 0.414 | 0.372 |

In Table 2, Table 3, and Table 5, we conducted experiments within a broader window size range of {336, 512, 720, 1600} to explore better results. Further implementation details can be found in the Appendix B.

## 5.3 RESULTS

**Multivariate Results** In multivariate long-term TS forecasting tasks, we compare `Conv` and `DConv` to the latest competitive Transformer-based and non-Transformer baselines on seven popular benchmarks. For fixed look-back window size of 512, Table 1 shows that our method can match or outperform the strongest model PatchTST on popular LTSF benchmarks. However, our model can further enhance predictive capabilities by utilizing a longer look-back window. In Table 2 and Table 5, from an overall perspective, our models achieve the best results in 86% cases in MSE and 86% cases in MAE for all compared models. Specifically, our models outperform the strongest Transformer baseline PatchTST in 86% cases in MSE and 75% cases in MAE. Additionally, our models perform best in 86% cases in MSE and 86% cases in MAE compared to the state-of-the-art MLP-based models (TiDE). Please note that our models have not performed effectively on the largest Traffic dataset. This limitation is due to the simplicity of our model, which comprises only a single convolution layer. We recognize the potential of stacking our Conv-based unit. By designing

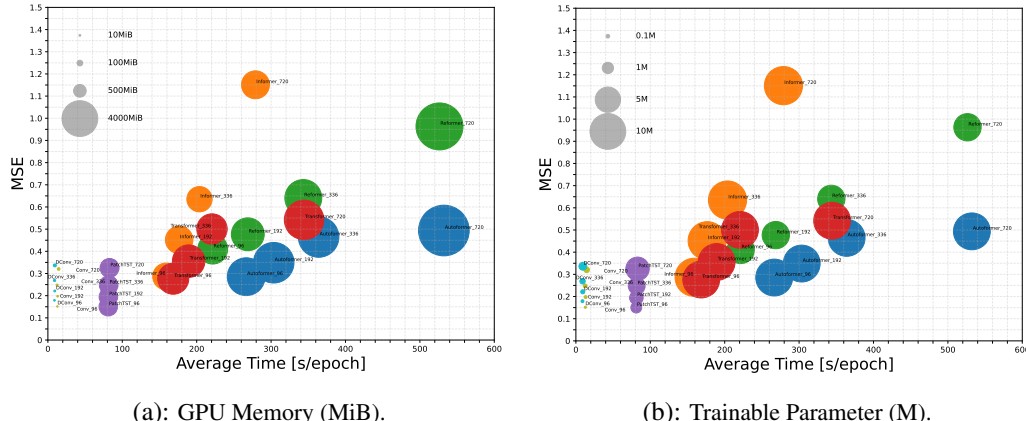

(a): GPU Memory (MiB).        (b): Trainable Parameter (M).

Figure 2: The ball chart illustrates the efficiency of various models with look-back window size 336 and forecasting lengths of {96, 192, 336, 720}, evaluated on the Weather dataset. Each ball's size is indicative of the model's complexity.

Table 3: Univariate long-term time series forecasting results. *Best* denotes re-implementation with a longer look-back window of {336, 512, 720, 1600}, consistently choosing the most optimal results.

| Methods | Metric | ETTm1 | | | | | ETTm2 | | | | | ETTh1 | | | | | ETTh2 | | | | |
|---|---|---|---|---|---|---|---|---|---|---|---|---|---|---|---|---|---|---|---|---|---|
| | | 96 | 192 | 336 | 720 | Avg | 96 | 192 | 336 | 720 | Avg | 96 | 192 | 336 | 720 | Avg | 96 | 192 | 336 | 720 | Avg |
| Conv-Best | MSE | **0.025** | 0.038 | 0.050 | **0.065** | 0.044 | 0.062 | 0.089 | 0.116 | 0.163 | 0.107 | 0.052 | 0.064 | 0.075 | 0.078 | 0.067 | 0.129 | 0.167 | 0.170 | 0.186 | 0.163 |
| | MAE | **0.120** | **0.147** | **0.169** | 0.196 | 0.158 | **0.181** | **0.223** | **0.257** | 0.316 | 0.244 | **0.174** | **0.197** | **0.217** | **0.228** | **0.204** | 0.278 | 0.325 | 0.338 | 0.351 | 0.323 |
| Dconv-Best | MSE | **0.025** | 0.037 | 0.049 | 0.066 | 0.044 | 0.062 | 0.089 | 0.116 | 0.162 | 0.107 | 0.053 | 0.069 | 0.077 | 0.079 | 0.069 | **0.128** | 0.168 | 0.175 | 0.199 | 0.167 |
| | MAE | **0.120** | **0.147** | 0.170 | 0.199 | 0.159 | 0.182 | **0.223** | **0.257** | 0.315 | 0.244 | 0.177 | 0.204 | 0.220 | 0.232 | 0.206 | 0.277 | **0.322** | 0.337 | 0.360 | 0.324 |
| PatchTST-Best | MSE | 0.026 | 0.039 | 0.053 | 0.073 | 0.048 | 0.065 | 0.093 | 0.120 | 0.171 | 0.112 | 0.055 | 0.071 | 0.076 | 0.087 | 0.072 | 0.129 | 0.168 | 0.171 | 0.223 | 0.173 |
| | MAE | 0.121 | 0.150 | 0.173 | 0.206 | 0.163 | 0.186 | 0.231 | 0.265 | 0.322 | 0.251 | 0.179 | 0.205 | 0.220 | 0.232 | 0.209 | 0.282 | 0.328 | **0.336** | 0.380 | 0.332 |
| TimesNet-Best | MSE | 0.029 | 0.047 | 0.059 | 0.077 | 0.053 | 0.066 | 0.113 | 0.133 | 0.183 | 0.124 | 0.056 | 0.072 | 0.081 | 0.082 | 0.073 | 0.136 | 0.186 | 0.197 | **0.172** | 0.173 |
| | MAE | 0.127 | 0.163 | 0.188 | 0.211 | 0.172 | 0.187 | 0.250 | 0.277 | 0.335 | 0.262 | 0.182 | 0.209 | 0.225 | 0.228 | 0.211 | 0.286 | 0.340 | 0.360 | **0.344** | 0.333 |
| MICN-Best | MSE | 0.033 | 0.050 | 0.065 | 0.089 | 0.059 | 0.065 | 0.105 | 0.132 | 0.166 | 0.117 | 0.063 | 0.075 | 0.092 | 0.129 | 0.090 | 0.125 | 0.181 | 0.203 | 0.272 | 0.195 |
| | MAE | 0.134 | 0.166 | 0.190 | 0.221 | 0.178 | 0.187 | 0.240 | 0.275 | 0.317 | 0.255 | 0.186 | 0.208 | 0.239 | 0.288 | 0.230 | **0.267** | 0.327 | 0.362 | 0.424 | 0.345 |

advanced architectures to enhance the model's capability to handle complex patterns and improve its performance on the larger Traffic dataset.

**Univariate Results**   We provide details about the univariate results of four typical ETT datasets. In Table 4, we compare some baselines under a look-back window size of 512. In Table 3, with an increase in input length, LTSF-Conv achieves the best results in 100% of cases for both average MSE and MAE metrics across various horizons. In particular, compared with the advanced Transformer-based solution PatchTST, our methods match or outperform it in all of the settings. Also, we observe that LTSF-Conv outperforms TimesNet by an avg. of 7.8%, and outperforms MICN by an avg. of 13.3%. For more exhaustive results, refer to the Appendix Table 4.

**Effciency Comparison**   In this section, we aim to illustrate that LTSF-Conv models exhibit significantly improved efficiency compared to Transformer-based models in terms of trainable param, GPU memory, running time, training time, and inference times. The ball charts presented in Figures 2(a) and 2(b) illustrate the MSE metric on the Weather dataset relative to the GPU memory and trainable parameter of the different models, where the size of each ball corresponds to the complexity of the model. Overall, It can be intuitively observed that Transformer-based models require approximately $13 \sim 67\times$ the number of parameters and $162 \sim 407\times$ the GPU memory compared to LTSF-Conv models, yet they exhibit higher mean squared error. More detailed results about the comparison of different datasets can be found in Table 13-15 of the Appendix. In particular, under a look-back length of 336 on the Electricity dataset, Conv achieves a 94% reduction ($2.21\rightarrow0.13$) in the number of training parameters required, a 99% reduction ($16032.05\rightarrow165.96$) in GPU memory usage, a 95% reduction ($265.65\rightarrow13.98$) in running time, and a 66% reduction ($56.74\rightarrow19.11$) in inference time compared to the advanced patchTST model.

**Interpretability**   LTSF-Conv is a set of convolution-based models, and the weights visualization within the projection layer can provide valuable insights into prediction. Here, let's take the Conv model as an example. We visualize the weights with two fixed input lengths and two different forecasting horizons. Specifically, Figure 3 illustrates the weight distribution of the Conv model

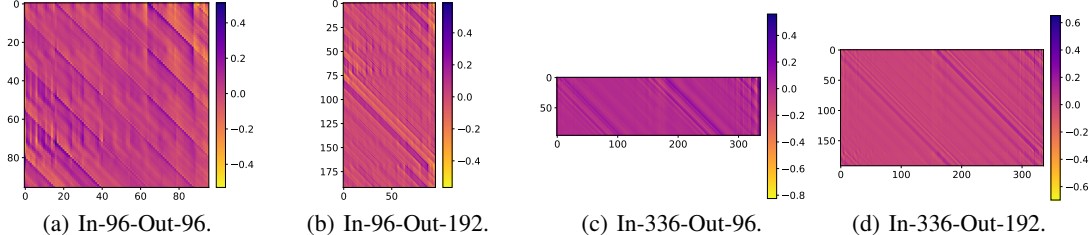

| (a) In-96-Out-96. | (b) In-96-Out-192. | (c) In-336-Out-96. | (d) In-336-Out-192. |

Figure 3: Visualization of the weights of Conv on Electricity benchmark. The X-axis represents the look-back window size, and the Y-axis represents the prediction length.

on the Electricity dataset. We can observe that the model assigns considerable weight to time steps with a 24-hour interval within the look-back window, indicating their higher relevance. Please note that the data is recorded at hourly intervals from 2012 to 2014, with each day comprising 24 time steps. This demonstrates that our model has the capability to capture a daily periodicity. Then we further expand the look-back window size to 336 which encompasses weekly patterns. Within these forecasting time steps, some time steps with a 168-hour interval within the look-back window are also assigned greater weight. This suggests that the Conv model is capable of capturing a weekly periodic pattern. More visualization results of DConv can be found in Figure 7 of the Appendix.

## 6 MORE ANALYSES ON LTSF-TRANSFORMER

In time series forecasting tasks, extending the forecasting horizon presents a practical but challenging problem. Although Transformer has achieved breakthrough success in some domains, applying it directly to long-term time series forecasting tasks remains a significant set of problems. Firstly, transformer-based models heavily rely on self-attention mechanisms to capture long-range dependencies and interactions within the data. However, the increased input and output sizes lead to a higher risk of overfitting, particularly in comparison to short-term forecasting scenarios. Secondly, the complex *encoder-decoder* structure of Transformers magnifies the memory bottleneck. In this section, we will explore the Transformer-based methods' suitability for long sequence prediction from two perspectives.

### 6.1 DO WE NEED AN ENCODER-DECODER STRUCTURE TO MODEL

Over the past few decades, there has been significant progress in artificial neural networks, driven by the belief that increasing network complexity can improve performance. These sophisticated networks are constructed with multiple layers comprising a large number of neurons or transformer blocks (Vaswani et al., 2017; Liu et al., 2021). Successful architectural design patterns for one task are often applied to address related tasks. Given the significant success in natural language processing, it is natural to use an *encoder-decoder* structure in LTSF tasks would be advantageous and imperative. However, as these models grow in complexity, they face challenges in optimization, and intricate operations, such as transformer models, call for a shift toward a simple structure. A recent paper has already questioned whether the *encoder-decoder* structure is effective for time series forecasting from a theoretical perspective (Liu et al., 2023). We performed experiments on nine real-world benchmarks to rigorously evaluate the performance and generalizability of the proposed conclusion. For a fair comparison, we chose three representative models with *encoder-decoder* structure and adopted the experimental settings used in the studies by (Wu et al., 2021). We maintained a consistent look-back length of 96 across all experiments. In the $Ori$ structure, both the encoder and decoder are kept intact, representing the original and complete architecture of the LTSF-Transformer models. In contrast, the $Half\text{-}Ex$ structure preserves the decoder as is, thereby reducing its complexity and capacity. By evaluating these two structures, our primary goal is to understand the influence of the structure reduction on the overall capabilities of the model. Table 16 of the Appendix presents a comparison of the performance between the complete $Encoder\text{-}Decoder$ structure and a reduced version comprising only half of the original components. As shown in the table, for the average performance across four different output horizons, the FEDformer-based reduced structure achieves the best results in 89% cases in terms of MSE and MAE metrics. The reduced structure based on Autoformer achieves the best results in 83% of the cases for both MSE and

MAE metrics. Similarly, the LogTrans-based reduced structure also exhibits strong performance, securing the best results in 94% of the cases for MSE and MAE metrics. Overall, by analyzing its performance across various metrics, we can gain valuable insights into the impact of reducing the *encoder-decoder* structure's complexity for the model's forecasting capabilities. More analysis of Transformers can be found in the Appendix C.1.

# 7  MORE ANALYSES ON LTSF-LINEAR

Recently, a series of LTSF-MLP models have emerged in the time series forecasting field. These models are known for their lightweight and fast nature, while still achieving performance comparable to or even better than Transformer-based methods. Although linear mapping can effectively learn periodic patterns in time series data, there are still some limitations compared to convolutional models for LTSF tasks. For the following subsection, we use the variable names: Linear-S: every channel shares the same linear layer, and Linear-CI: a linear layer for each channel individually (Channel Independent).

## 7.1  LIMITATIONS OF DEALING WITH MULTIPLE PERIODS AMONG CHANNELS

In the context of LTSF tasks, linear mapping proves effective in capturing and modeling periodicity in univariate data (Li et al., 2023). However, when dealing with tasks involving multiple periods across channels, where each channel represents a distinct aspect or feature of the data, the limitations of the Linear-S mapping become apparent. It lacks the capacity to capture intricate interactions between channels, making it challenging to model complex patterns and interactions accurately. Furthermore, LTSF tasks require capturing dependencies over extended periods. With multiple channels, the complexity of capturing long-term dependencies increases further. As a result, the forecasting accuracy of the model may be limited. Figure 6 of the Appendix shows the forecasting results of different Linear-S models applied to the real-world dataset consisting of distinct periodic channels. The results indicate that the proposed `Conv` basic unit is capable of capturing the underlying curves, whereas Linear-S struggles to fit the multiple channels data more effectively. Specifically, our model demonstrates accuracy in predicting the peaks and troughs at the early stages, while also achieving impressive precision in forecasting the trend in the long-term future. However, the Linear-S model exhibits an over-smoothing problem, causing the predicted trends to deviate further from the actual data trends. As the prediction length increases, this trend deviation becomes even more pronounced, highlighting the limitations of the model's ability to capture the true dynamics of the data.

To address this challenge, a potential solution is to use Linear-CI modeling. However, this approach does come with a trade-off, as it noticeably increases the computational overhead required for the training process. Table 17 of the Appendix measures the computational cost of each Linear-CI model on the Electricity benchmark. It can be observed that the utilization of Linear-CI models leads to a substantial increase in the number of parameters, memory usage (Max), and running time (Avg) compared to Linear-S models. Specifically, RLinear-CI leads to a substantial increase in the average number of trainable parameters (by $\sim 320\times$), maximum GPU memory consumption (by $\sim 2\times$), and epoch time (by $\sim 7\times$). Similarly, DLinear-CI results in a significant rise in the average number of trainable parameters (by $\sim 320\times$), maximum GPU memory usage (by $\sim 3\times$), and running time (by $\sim 5\times$). This increase is primarily due to the number of channels. Notably, when the lookback window size is increased, some datasets with a large number of channels, such as Traffic (862 channels), encounter an unacceptable training time.

# 8  CONCLUSION

This work systematically investigates the limitations of Transformer-based and MLP-based solutions in long-term time series forecasting tasks. We use an embarrassingly simple convolution unit as a temporal feature extractor to verify our claims. By embracing the depthwise convolution mechanism, the `LTSF-Conv` models exhibit their adeptness in achieving a balance between computational efficiency and model performance. This progress is particularly valuable in situations involving the processing of long sequences, as it adeptly tackles the computational limitations inherent in conventional Transformers and MLPs. Note that LTSF-Conv models serve as straightforward yet competitive basic units, exhibiting promising potential for further expansion.

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

Table 4: Univariate long-term time series forecasting results on ETT full benchmark, with a fixed input length sl=512.

| Methods | | Conv (Ours) | | DConv (Ours) | | PatchTST 2023 | | TimesNet 2023 | | MICN-regre 2023 | | FEDformer 2022 | | Autoformer 2022 | | DLinear 2022 | | Informer 2021 | | LogTrans 2019 | |
|---|---|---|---|---|---|---|---|---|---|---|---|---|---|---|---|---|---|---|---|---|---|
| Metric | | MSE | MAE | MSE | MAE | MSE | MAE | MSE | MAE | MSE | MAE | MSE | MAE | MSE | MAE | MSE | MAE | MSE | MAE | MSE | MAE |
| ETTm1 | 96 | **0.026** | **0.122** | **0.026** | **0.122** | **0.026** | 0.123 | 0.031 | 0.135 | 0.036 | 0.142 | 0.033 | 0.140 | 0.056 | 0.183 | 0.028 | 0.123 | 0.109 | 0.277 | 0.049 | 0.171 |
| | 192 | **0.039** | **0.150** | **0.039** | **0.150** | 0.040 | 0.151 | 0.048 | 0.169 | 0.052 | 0.171 | 0.058 | 0.186 | 0.081 | 0.216 | 0.045 | 0.156 | 0.151 | 0.310 | 0.157 | 0.317 |
| | 336 | **0.052** | **0.173** | **0.052** | **0.173** | 0.053 | 0.174 | 0.059 | 0.188 | 0.070 | 0.202 | 0.084 | 0.231 | 0.076 | 0.218 | 0.061 | 0.182 | 0.427 | 0.591 | 0.289 | 0.459 |
| | 720 | 0.072 | **0.204** | **0.070** | **0.204** | 0.073 | 0.206 | 0.077 | 0.211 | 0.089 | 0.221 | 0.102 | 0.250 | 0.110 | 0.267 | 0.080 | 0.210 | 0.438 | 0.586 | 0.430 | 0.579 |
| | Avg | **0.047** | **0.162** | **0.047** | **0.162** | 0.048 | 0.163 | 0.054 | 0.176 | 0.062 | 0.184 | 0.069 | 0.201 | 0.080 | 0.221 | 0.053 | 0.167 | 0.281 | 0.441 | 0.231 | 0.381 |
| ETTm2 | 96 | **0.063** | **0.182** | **0.063** | 0.183 | 0.065 | 0.187 | 0.078 | 0.212 | 0.077 | 0.208 | 0.067 | 0.198 | 0.065 | 0.189 | **0.063** | 0.183 | 0.088 | 0.225 | 0.075 | 0.208 |
| | 192 | 0.091 | **0.225** | **0.090** | **0.225** | 0.093 | 0.231 | 0.108 | 0.252 | 0.093 | 0.229 | 0.102 | 0.245 | 0.118 | 0.256 | 0.092 | 0.227 | 0.132 | 0.283 | 0.129 | 0.275 |
| | 336 | 0.119 | **0.261** | **0.118** | **0.261** | 0.121 | 0.266 | 0.138 | 0.289 | 0.132 | 0.278 | 0.130 | 0.279 | 0.154 | 0.305 | 0.119 | 0.261 | 0.180 | 0.336 | 0.154 | 0.302 |
| | 720 | 0.172 | 0.321 | 0.171 | **0.320** | 0.172 | 0.322 | 0.183 | 0.335 | 0.183 | 0.335 | **0.166** | **0.317** | 0.178 | 0.325 | 0.182 | 0.335 | 0.300 | 0.435 | 0.160 | 0.321 |
| | Avg | **0.111** | **0.247** | **0.111** | **0.247** | 0.112 | 0.251 | 0.127 | 0.272 | 0.117 | 0.258 | 0.119 | 0.261 | 0.129 | 0.271 | 0.112 | 0.247 | 0.175 | 0.319 | 0.129 | 0.276 |
| ETTh1 | 96 | **0.053** | **0.177** | 0.056 | 0.183 | 0.059 | 0.189 | 0.063 | 0.196 | 0.066 | 0.198 | 0.079 | 0.215 | 0.071 | 0.206 | 0.056 | 0.180 | 0.193 | 0.377 | 0.283 | 0.468 |
| | 192 | **0.065** | **0.198** | 0.072 | 0.212 | 0.074 | 0.215 | 0.079 | 0.224 | 0.100 | 0.248 | 0.104 | 0.245 | 0.114 | 0.262 | 0.071 | 0.204 | 0.217 | 0.395 | 0.234 | 0.409 |
| | 336 | **0.075** | **0.217** | 0.077 | 0.221 | 0.076 | 0.220 | 0.081 | 0.225 | 0.115 | 0.269 | 0.119 | 0.270 | 0.107 | 0.258 | 0.098 | 0.244 | 0.202 | 0.381 | 0.386 | 0.546 |
| | 720 | **0.082** | **0.227** | 0.082 | 0.229 | 0.082 | 0.228 | **0.082** | 0.228 | 0.135 | 0.424 | 0.142 | 0.299 | 0.126 | 0.283 | 0.189 | 0.359 | 0.183 | 0.355 | 0.475 | 0.629 |
| | Avg | **0.069** | **0.205** | 0.071 | 0.211 | 0.074 | 0.215 | 0.076 | 0.218 | 0.135 | 0.285 | 0.111 | 0.257 | 0.104 | 0.252 | 0.103 | 0.246 | 0.198 | 0.377 | 0.344 | 0.513 |
| ETTh2 | 96 | 0.135 | 0.285 | 0.135 | 0.285 | 0.131 | 0.284 | 0.146 | 0.300 | 0.146 | 0.299 | **0.128** | **0.271** | 0.153 | 0.306 | 0.131 | 0.279 | 0.213 | 0.373 | 0.217 | 0.379 |
| | 192 | 0.173 | 0.328 | 0.173 | **0.328** | **0.171** | 0.329 | 0.195 | 0.351 | 0.184 | 0.338 | 0.185 | 0.330 | 0.204 | 0.351 | 0.176 | 0.329 | 0.227 | 0.387 | 0.281 | 0.429 |
| | 336 | 0.179 | 0.342 | 0.177 | 0.340 | **0.171** | **0.336** | 0.197 | 0.360 | 0.203 | 0.362 | 0.231 | 0.378 | 0.246 | 0.389 | 0.209 | 0.367 | 0.242 | 0.401 | 0.293 | 0.437 |
| | 720 | 0.219 | 0.376 | 0.219 | 0.375 | 0.223 | 0.380 | **0.172** | **0.344** | 0.410 | 0.524 | 0.278 | 0.420 | 0.268 | 0.409 | 0.276 | 0.426 | 0.291 | 0.439 | 0.218 | 0.387 |
| | Avg | 0.177 | 0.333 | 0.176 | **0.332** | **0.174** | **0.332** | 0.178 | 0.339 | 0.236 | 0.381 | 0.205 | 0.349 | 0.217 | 0.363 | 0.198 | 0.350 | 0.243 | 0.400 | 0.252 | 0.408 |

Table 5: Multivariate long-term time series forecasting best results.

| Methods | Metric | ETTh1 | | | | | ETTh2 | | | | | Traffic | | | | |
|---|---|---|---|---|---|---|---|---|---|---|---|---|---|---|---|---|
| | | 96 | 192 | 336 | 720 | Avg | 96 | 192 | 336 | 720 | Avg | 96 | 192 | 336 | 720 | Avg |
| Conv-Best | MSE | **0.365** | 0.401 | 0.424 | 0.450 | 0.411 | **0.268** | 0.327 | 0.329 | **0.379** | 0.326 | 0.383 | 0.397 | 0.411 | 0.450 | 0.410 |
| | MAE | 0.393 | 0.416 | 0.428 | 0.460 | 0.425 | 0.339 | 0.382 | 0.390 | 0.424 | 0.384 | 0.271 | 0.275 | 0.282 | 0.302 | 0.283 |
| Dconv-Best | MSE | 0.366 | **0.400** | 0.421 | 0.429 | **0.404** | 0.269 | 0.326 | 0.321 | 0.382 | 0.325 | 0.378 | 0.390 | 0.404 | 0.442 | 0.404 |
| | MAE | **0.382** | 0.412 | 0.422 | 0.446 | **0.416** | 0.334 | 0.375 | 0.386 | 0.428 | 0.381 | 0.264 | 0.269 | 0.275 | 0.294 | 0.276 |
| PatchTST-Best | MSE | 0.370 | 0.414 | 0.422 | 0.447 | 0.413 | 0.274 | 0.339 | 0.329 | 0.379 | 0.330 | 0.360 | 0.379 | 0.392 | 0.432 | 0.390 |
| | MAE | 0.400 | 0.421 | 0.440 | 0.468 | 0.432 | 0.336 | 0.379 | 0.384 | 0.422 | 0.380 | 0.249 | 0.256 | 0.264 | 0.286 | 0.263 |
| Preformer-Best | MSE | 0.438 | 0.457 | 0.464 | 0.503 | 0.466 | 0.339 | 0.385 | 0.394 | 0.466 | 0.396 | 0.560 | 0.565 | 0.577 | 0.597 | 0.575 |
| | MAE | 0.455 | 0.474 | 0.481 | 0.512 | 0.481 | 0.384 | 0.435 | 0.441 | 0.479 | 0.435 | 0.349 | 0.349 | 0.351 | 0.358 | 0.352 |
| TiDE | MSE | 0.375 | 0.412 | 0.435 | 0.454 | 0.419 | 0.270 | 0.332 | 0.360 | 0.419 | 0.345 | 0.336 | 0.346 | 0.355 | 0.386 | **0.355** |
| | MAE | 0.398 | 0.422 | 0.433 | 0.465 | 0.429 | 0.336 | 0.380 | 0.407 | 0.451 | 0.393 | 0.253 | 0.257 | 0.260 | 0.273 | **0.260** |
| MPPN | MSE | 0.371 | 0.405 | 0.426 | 0.436 | 0.409 | 0.278 | 0.344 | 0.362 | 0.393 | 0.344 | 0.387 | 0.396 | 0.410 | 0.449 | 0.410 |
| | MAE | 0.393 | 0.413 | 0.425 | 0.452 | 0.420 | 0.335 | 0.380 | 0.400 | 0.434 | 0.387 | 0.271 | 0.273 | 0.279 | 0.301 | 0.281 |

## A PROOFS

**Corollary 1.** *When considering transformed periodic sequences $x(t) = a \cdot x(t - p) + c$, the convolutional model still has an explicit solution to Equation 2 as*

$$w(i) = \begin{cases} a, & \text{if } i = (sl - \alpha \cdot p) \bmod p \\ 0, & \text{otherwise} \end{cases}, \quad b_i = c. \tag{8}$$

We have discovered that a single convolutional network can successfully capture periodicity in a time series. However, a general time series can be decomposed into two components: a periodic sequence and a sequence with a smooth trend (Wu et al., 2021) (Zhou et al., 2022).

**Theorem 2.** *Let $x(t) = s(t) + g(t)$, where $s(t)$ represents the seasonal signal with period $p$ and $g(t)$ satisfies $V$-Lipschitz smooth (i.e. $|g(a) - g(b)| \le V|a - b|$). Then there exists a convolutional model with look-back window length $sl = p + \epsilon, \epsilon \ge 0$, such that $|x[sl + h] - \hat{x}[sl + h]| \le V \cdot (p + \epsilon), h = 0, \ldots, H - 1$.*

*Proof.* We denote the length of historical data as $sl$. In terms of the ground truth of the future time series, it can be described as follows:

$$x[sl + h] = s(p + \epsilon + h) + g(p + \epsilon + h) \tag{9}$$

Assuming that the convolutional network is only capable of capturing periodic patterns, we can use Equation 3 directly as an approximate solution. Then the $h$-$th$ true value will be predicted:

$$\begin{aligned} \hat{x}[sl + h] &= x(h + (p + \epsilon - \alpha \cdot p) \bmod p) \\ &= s(h + (p + \epsilon) \bmod p) + g(h + (p + \epsilon - \alpha \cdot p) \bmod p) \\ &= s(p + \epsilon + h) + g((p + \epsilon - \alpha \cdot p) \bmod p + h) \end{aligned} \tag{10}$$

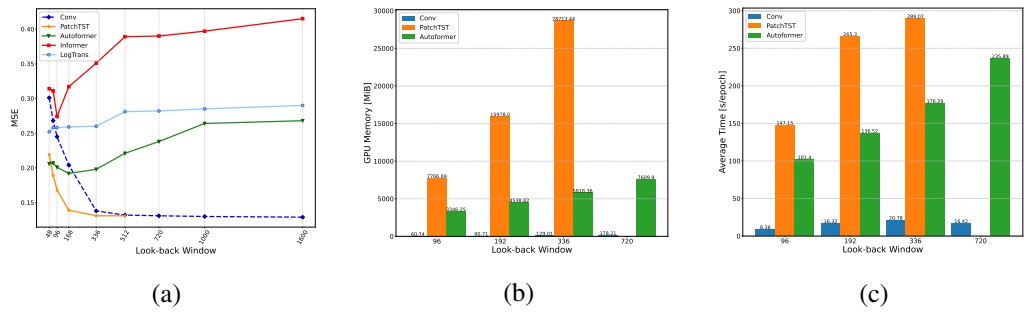

(a)                                   (b)                                   (c)

Figure 4: The MSE, GPU memory, and average running time (Y-axis) of models under different look-back window sizes (X-axis) on the Electricity benchmark (T=96).

So we can get

$$
\begin{aligned}
|x[sl+h] - \hat{x}[sl+h]| &= g(p+\epsilon+h) - g((p+\epsilon-\alpha \cdot p) \bmod p + h) \\
&\leq V \cdot (p+\epsilon - (p+\epsilon-\alpha \cdot p) \bmod p) \\
&\leq V \cdot (p+\epsilon)
\end{aligned}
\tag{11}
$$

This analysis highlights the associated error is bounded given an appropriate look-back window size.

## B  IMPLEMENTATION DETAILS

All experiments are conducted on the NVIDIA GeForce RTX 2080 Ti GPU, implemented in Py-Torch. The training process is early stopped within five epochs. We use the ADAM optimizer with a decay learning rate schedule, which gradually decreases the learning rate over time in a smooth and controlled manner. The initial learning rate is set to 5·1e-3. and the hyper-parameter convolution kernel size is fine-tuned within the range {20, 35, 55, 128, 196}. Under the configuration with a look-back window size of 512, the specific hyper-parameters chosen of the Conv model for each dataset are summarized in Table 7 and Table 8. For long-term time series forecasting, all best experimental results are derived from three runs with different random seeds, and then calculating the mean. The related standard deviation and mean values results are summarized in Table 9.

## C  MORE EXPERIMENTAL RESULTS

Table 6: Summary of seven benchmarks.

| Datasets | Weather | Electricity | Traffic | ETTm1 | ETTm2 | ETTh1 | ETTh2 |
|---|---|---|---|---|---|---|---|
| Features | 21 | 321 | 862 | 7 | 7 | 7 | 7 |
| Timesteps | 52,696 | 26,304 | 17,544 | 69,680 | 69,680 | 17,420 | 17,420 |
| Granularity | 10 Minutes | 1 Hour | 1 Hour | 15 Minutes | 15 Minutes | 1 Hour | 1 Hour |
| Date | 2020/1-2021/1 | 2016/7-2019/7 | 2016/7-2018/7 | 2016/7-2018/6 | 2016/7-2018/6 | 2016/7-2018/6 | 2016/7-2018/6 |

### C.1  CAN EXISTING LTSF TRANSFORMERS BENEFIT FROM LONGER LOOK-BACK WINDOWS

For the LTSF tasks, a longer look-back window expands the receptive field, which can potentially improve the predictive accuracy of forecasting models. Here, we compare `Conv` and Transformer-based solutions across various look-back window sizes on the Electricity benchmark. We conducted experiments using a range of look-back window sizes {48, 72, 96, 168, 336, 512, 720, 1000, 1600} for long-term forecasting, where T=96. In Figure 4 (a), it is evident that the performance of existing Transformer-based models either degrades or remains stable as the look-back window size increases. These models struggle to efficiently extract more valuable temporal information when extending the input horizon. This finding contradicts our initial hypothesis. However, as the look-back window size increases, `Conv` demonstrates significant improvements in performance, highlighting its capacity to effectively learn from extended historical data.

Table 7: Hyperparameters of multivariate long-term time series forecasting tasks. The look-back window size is 512.

| Dataset | Pre_len | Batch_Size | Learning_Rate | Kernel_Size | Individual |
|---------|---------|------------|---------------|-------------|------------|
| ETTh1 | 96 | 16 | 0.005 | 55 | 0 |
| | 192 | 16 | 0.005 | 55 | 0 |
| | 336 | 128 | 0.005 | 78 | 1 |
| | 720 | 16 | 0.005 | 55 | 0 |
| ETTh2 | 96 | 128 | 0.005 | 55 | 0 |
| | 192 | 128 | 0.005 | 78 | 0 |
| | 336 | 128 | 0.0005 | 78 | 0 |
| | 720 | 128 | 0.0001 | 78 | 0 |
| ETTm1 | 96 | 16 | 0.005 | 78 | 1 |
| | 192 | 16 | 0.005 | 35 | 0 |
| | 336 | 16 | 0.005 | 35 | 0 |
| | 720 | 16 | 0.005 | 35 | 0 |
| ETTm2 | 96 | 16 | 0.005 | 24 | 1 |
| | 192 | 16 | 0.005 | 55 | 1 |
| | 336 | 16 | 0.005 | 35 | 0 |
| | 720 | 64 | 0.005 | 24 | 0 |
| Weather | 96 | 16 | 0.005 | 55 | 0 |
| | 192 | 16 | 0.005 | 78 | 1 |
| | 336 | 16 | 0.005 | 24 | 1 |
| | 720 | 128 | 0.005 | 78 | 0 |
| Electricity | 96 | 16 | 0.005 | 55 | 0 |
| | 192 | 128 | 0.005 | 55 | 1 |
| | 336 | 64 | 0.005 | 55 | 0 |
| | 720 | 16 | 0.005 | 55 | 0 |
| Traffic | 96 | 16 | 0.005 | 35 | 0 |
| | 192 | 16 | 0.005 | 35 | 0 |
| | 336 | 16 | 0.005 | 24 | 0 |
| | 720 | 16 | 0.005 | 24 | 0 |

Table 8: Hyperparameters of univariate long-term time series forecasting tasks. The look-back window size is 512.

| Dataset | Pre_Len | Batch_size | Learning_rate | Kernel_Size | Individual |
|---------|---------|------------|---------------|-------------|------------|
| ETTh1 | 96 | 16 | 0.005 | 24 | 0 |
| | 192 | 16 | 0.005 | 55 | 0 |
| | 336 | 64 | 0.005 | 55 | 0 |
| | 720 | 128 | 0.005 | 78 | 0 |
| ETTh2 | 96 | 128 | 0.005 | 24 | 0 |
| | 192 | 16 | 0.005 | 35 | 0 |
| | 336 | 128 | 0.005 | 24 | 0 |
| | 720 | 128 | 0.005 | 24 | 0 |
| ETTm1 | 96 | 64 | 0.005 | 55 | 0 |
| | 192 | 64 | 0.005 | 24 | 0 |
| | 336 | 16 | 0.005 | 35 | 0 |
| | 720 | 16 | 0.005 | 35 | 0 |
| ETTm2 | 96 | 16 | 0.005 | 55 | 0 |
| | 192 | 16 | 0.005 | 24 | 0 |
| | 336 | 64 | 0.005 | 24 | 0 |
| | 720 | 16 | 0.005 | 24 | 0 |
| Weather | 96 | 64 | 0.005 | 78 | 0 |
| | 192 | 16 | 0.005 | 24 | 0 |
| | 336 | 16 | 0.005 | 78 | 0 |
| | 720 | 128 | 0.005 | 55 | 0 |
| Electricity | 96 | 16 | 0.005 | 35 | 0 |
| | 192 | 64 | 0.005 | 35 | 0 |
| | 336 | 64 | 0.005 | 35 | 0 |
| | 720 | 16 | 0.005 | 35 | 0 |
| Traffic | 96 | 16 | 0.005 | 24 | 0 |
| | 192 | 64 | 0.005 | 35 | 0 |
| | 336 | 16 | 0.005 | 35 | 0 |
| | 720 | 16 | 0.005 | 24 | 0 |

In Figure 4 (b) and (c), we illustrate the GPU memory utilization and average running time of Transformer-based models and `Conv` model as the look-back window size is varied from 48 to 1000. Notably, PatchTST and Autoformer exhibit heightened sensitivity to changes in window size. Moreover, PatchTST runs out of memory when the look-back window size is greater than or equal to 720. In contrast, our model maintains relatively stable memory utilization while the lookback window increases. `Conv` achieves superior accuracy while concurrently sustaining significantly improved memory efficiency and reduced average running time.

Table 9: The error bars of Conv-Best with 3 runs, output length $H \in \{96, 192, 336, 720\}$ on Traffic, Electricity, and ETTm2.

| Dataset | | Traffic | | | | | Electricity | | | | | ETTm2 | | | | |
|---|---|---|---|---|---|---|---|---|---|---|---|---|---|---|---|---|
| Metric | | Seed1 | Seed2 | Seed3 | Mean | Std. | Seed1 | Seed2 | Seed3 | Mean | Std. | Seed1 | Seed2 | Seed3 | Mean | Std. |
| 96 | MSE | 0.3837 | 0.3838 | 0.3833 | 0.3836 | 0.00021 | 0.1294 | 0.1294 | 0.1295 | 0.1295 | 0.00005 | 0.1615 | 0.1621 | 0.1618 | 0.1618 | 0.00023 |
| | MAE | 0.2714 | 0.2715 | 0.2708 | 0.2712 | 0.00029 | 0.2249 | 0.2249 | 0.2250 | 0.2250 | 0.00005 | 0.2492 | 0.2494 | 0.2495 | 0.2494 | 0.00012 |
| 192 | MSE | 0.3977 | 0.3970 | 0.3970 | 0.3972 | 0.00035 | 0.1438 | 0.1441 | 0.1437 | 0.1439 | 0.00015 | 0.2174 | 0.2171 | 0.2171 | 0.2172 | 0.00014 |
| | MAE | 0.2755 | 0.2751 | 0.2752 | 0.2753 | 0.00018 | 0.2385 | 0.2390 | 0.2385 | 0.2387 | 0.00024 | 0.2879 | 0.2878 | 0.2878 | 0.2878 | 0.00006 |
| 336 | MSE | 0.4114 | 0.4110 | 0.4110 | 0.4111 | 0.00022 | 0.1594 | 0.1594 | 0.1598 | 0.1595 | 0.00019 | 0.2576 | 0.2577 | 0.2575 | 0.2576 | 0.00008 |
| | MAE | 0.2820 | 0.2819 | 0.2820 | 0.2820 | 0.00005 | 0.2549 | 0.2553 | 0.2553 | 0.2551 | 0.00018 | 0.3292 | 0.3294 | 0.3292 | 0.3293 | 0.00007 |
| 720 | MSE | 0.4509 | 0.4499 | 0.4507 | 0.4505 | 0.00041 | 0.1955 | 0.1949 | 0.1950 | 0.1951 | 0.00023 | 0.3230 | 0.3268 | 0.3243 | 0.3247 | 0.00157 |
| | MAE | 0.3027 | 0.3017 | 0.3019 | 0.3021 | 0.00042 | 0.2863 | 0.2861 | 0.2863 | 0.2863 | 0.00008 | 0.3787 | 0.3794 | 0.3796 | 0.3793 | 0.00038 |

## C.2 ABLATION STUDIES

In this section, we conduct additional ablation experiments to verify the validity of the group-wise convolution (the number of channels is equal to the variable dimension and the number of filters) and cross-channel convolution in the `Conv` model. Tables 10 and 11 display quantitative results with fixed input lengths of 336 and 512 and four different forecasting horizons. We use the following variable names: ① `Conv` uses depthwise convolution (channel independence), ② `Conv`$_{CD}$ uses so-called channel-dependent convolution. It proves that the group-wise convolution can better mine the long sequence information, at least for existing benchmarks.

Table 10: Ablations on depthwise convolution with a look-back window size of 336 on six benchmarks.

| Benchmark | | Conv | | | | Conv$_{CD}$ | | | |
|---|---|---|---|---|---|---|---|---|---|
| | | 96 | 192 | 336 | 720 | 96 | 192 | 336 | 720 |
| ETTh1 | MSE | 0.370 | 0.407 | 0.422 | 0.451 | 0.394 | 0.475 | 0.464 | 0.535 |
| | MAE | 0.394 | 0.417 | 0.428 | 0.458 | 0.418 | 0.461 | 0.465 | 0.509 |
| ETTh2 | MSE | 0.277 | 0.341 | 0.330 | 0.379 | 0.340 | 0.398 | 0.373 | 0.452 |
| | MAE | 0.343 | 0.383 | 0.387 | 0.420 | 0.394 | 0.433 | 0.421 | 0.466 |
| ETTm1 | MSE | 0.287 | 0.328 | 0.366 | 0.422 | 0.305 | 0.346 | 0.378 | 0.430 |
| | MAE | 0.334 | 0.358 | 0.380 | 0.413 | 0.352 | 0.375 | 0.390 | 0.421 |
| ETTm2 | MSE | 0.163 | 0.217 | 0.271 | 0.368 | 0.175 | 0.247 | 0.330 | 0.437 |
| | MAE | 0.250 | 0.288 | 0.324 | 0.385 | 0.264 | 0.310 | 0.368 | 0.426 |
| Weather | MSE | 0.143 | 0.185 | 0.237 | 0.313 | 0.145 | 0.192 | 0.247 | 0.321 |
| | MAE | 0.189 | 0.230 | 0.271 | 0.326 | 0.199 | 0.243 | 0.284 | 0.333 |
| Electricity | MSE | 0.136 | 0.150 | 0.167 | 0.204 | 0.250 | 0.279 | 0.274 | 0.310 |
| | MAE | 0.230 | 0.243 | 0.259 | 0.292 | 0.356 | 0.371 | 0.372 | 0.396 |

Table 11: Ablations on depthwise convolution with a look-back window size of 512 on six benchmarks.

| Benchmark | | Conv | | | | Conv$_{CD}$ | | | |
|---|---|---|---|---|---|---|---|---|---|
| | | 96 | 192 | 336 | 720 | 96 | 192 | 336 | 720 |
| ETTh1 | MSE | 0.365 | 0.401 | 0.419 | 0.464 | 0.450 | 0.441 | 0.512 | 0.605 |
| | MAE | 0.393 | 0.416 | 0.437 | 0.472 | 0.455 | 0.460 | 0.501 | 0.541 |
| ETTh2 | MSE | 0.269 | 0.329 | 0.335 | 0.379 | 0.336 | 0.371 | 0.369 | 0.470 |
| | MAE | 0.339 | 0.383 | 0.394 | 0.424 | 0.385 | 0.411 | 0.419 | 0.481 |
| ETTm1 | MSE | 0.292 | 0.332 | 0.364 | 0.418 | 0.315 | 0.358 | 0.390 | 0.445 |
| | MAE | 0.338 | 0.361 | 0.380 | 0.411 | 0.359 | 0.385 | 0.405 | 0.441 |
| ETTm2 | MSE | 0.161 | 0.216 | 0.271 | 0.361 | 0.179 | 0.248 | 0.311 | 0.430 |
| | MAE | 0.249 | 0.288 | 0.327 | 0.387 | 0.272 | 0.318 | 0.363 | 0.429 |
| Weather | MSE | 0.140 | 0.183 | 0.234 | 0.306 | 0.150 | 0.199 | 0.246 | 0.315 |
| | MAE | 0.190 | 0.230 | 0.271 | 0.325 | 0.203 | 0.251 | 0.283 | 0.334 |
| Electricity | MSE | 0.132 | 0.145 | 0.161 | 0.201 | 0.252 | 0.246 | 0.263 | 0.282 |
| | MAE | 0.227 | 0.241 | 0.257 | 0.289 | 0.360 | 0.351 | 0.367 | 0.378 |

## C.3 ROBUSTNESS EXPERIMENTS

In order to assess the potential influence of different random initializations for long-term forecasting, we conduct experiments on Weather, Electricity, Traffic, ETTm2, ETTh1, and ETTh2 datasets. Each experimental configuration was repeated with five random seeds. The robust experimental results are provided in Figure 5. In general, we observe that `DConv` exhibits reliability and stability towards different initialization. However, on the traffic dataset, there is a certain degree of fluctuation in the results. We think the reason for the poor performance of `DConv` on traffic datasets is excessively high variable dimensionality ($C = 862$). A simple two-layer network is insufficient for modeling high-dimensional and large-scale datasets due to the limited parameter capacity of the model.

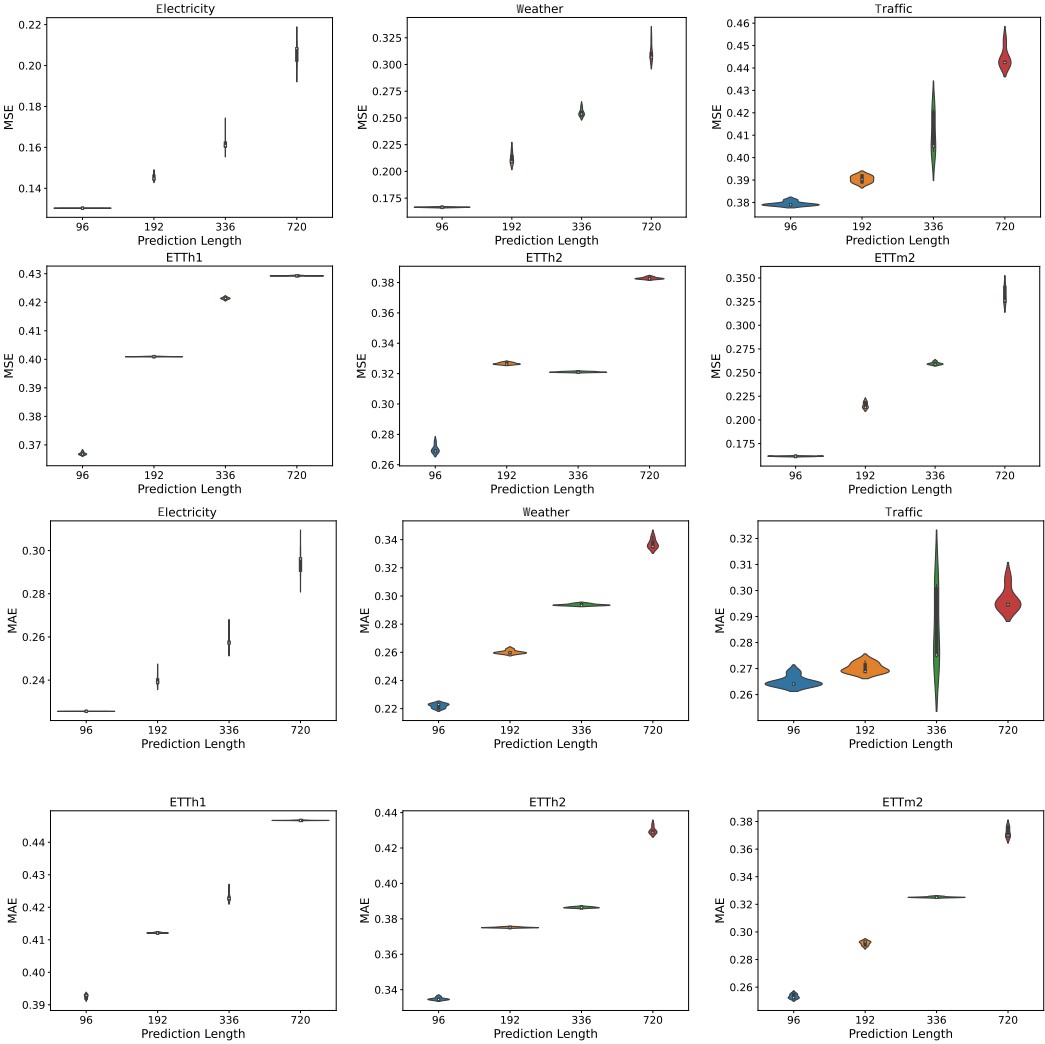

Figure 5: Robust experiments of DConv with different random seeds. The X-axis corresponds to the prediction length, and the Y-axis represents the MSE/MAE metrics.

## C.4 MORE EXPERIMENTAL DATASETS

In this section, we add additional experiments to further verify the validity of LTSF-Conv model in different LTSF tasks. Solar power prediction is a crucial aspect within the realm of renewable energy, wielding significant influence across diverse domains. We add two datasets from real-world applications, namely the Solar-Jinta and Solar-Alabama benchmarks. Solar-Jinta records seven key

Table 12: Multivariate long-term time series forecasting results on Solar benchmarks.

| Benchmarks | Methods | Metric | | | | | | | | | |
|---|---|---|---|---|---|---|---|---|---|---|---|
| | | MSE | | | | | MAE | | | | |
| | | 96 | 192 | 336 | 720 | Avg | 96 | 192 | 336 | 720 | Avg |
| Solar-Energy | DConv | **0.173** | **0.189** | **0.201** | **0.206** | **0.192** | **0.229** | **0.237** | **0.244** | **0.245** | **0.239** |
| | PatchTST | 0.178 | 0.192 | 0.203 | 0.218 | 0.197 | 0.248 | 0.265 | 0.267 | 0.274 | 0.263 |
| | Flowformer | 0.190 | 0.267 | 0.279 | 0.243 | 0.245 | 0.229 | 0.263 | 0.268 | 0.252 | 0.253 |
| | Reformer | 0.195 | 0.223 | 0.253 | 0.281 | 0.238 | 0.234 | 0.252 | 0.286 | 0.286 | 0.265 |
| | Informer | 0.195 | 0.220 | 0.259 | 0.246 | 0.230 | 0.241 | 0.241 | 0.274 | 0.266 | 0.256 |
| | LogTrans | 0.219 | 0.217 | 0.224 | 0.241 | 0.225 | 0.240 | 0.244 | 0.258 | 0.269 | 0.252 |
| | Dlinear | 0.221 | 0.231 | 0.247 | 0.255 | 0.239 | 0.294 | 0.301 | 0.317 | 0.314 | 0.307 |
| Solar- Jinta | DConv | **0.480** | **0.521** | **0.539** | **0.648** | **0.547** | 0.481 | **0.496** | **0.522** | **0.573** | **0.518** |
| | PatchTST | 0.491 | 0.602 | 0.617 | 0.710 | 0.605 | **0.464** | 0.523 | 0.533 | 0.601 | 0.530 |
| | Flowformer | 0.646 | 0.792 | 0.748 | 1.065 | 0.812 | 0.565 | 0.624 | 0.611 | 0.783 | 0.645 |
| | Reformer | 0.954 | 0.934 | 0.993 | 0.995 | 0.969 | 0.773 | 0.757 | 0.742 | 0.730 | 0.751 |
| | Informer | 0.718 | 0.821 | 0.847 | 1.102 | 0.872 | 0.616 | 0.656 | 0.737 | 0.855 | 0.716 |
| | LogTrans | 0.704 | 0.817 | 0.754 | 1.012 | 0.821 | 0.594 | 0.651 | 0.642 | 0.762 | 0.662 |
| | Dlinear | 0.523 | 0.588 | 0.638 | 0.734 | 0.620 | 0.496 | 0.531 | 0.562 | 0.616 | 0.551 |

meteorological factors of solar radiation, collected by the hour. Solar-Alabama [1] contains the solar power production of 137 PV plants in the USA, with a data granularity of 10 minutes. The solar power of different PV plants is influenced by varying geographical and weather conditions. Table 12 provides a summary of prediction results from several popular baselines on the Solar-Energy datasets. It can be observed that DConv outperforms the other baselines for most horizons by a large margin. Moreover, compared to other datasets, the Solar-Jinta benchmark has a smaller size. The experimental results also indicate that our model performs equally well on small datasets, ensuring its generalizability and robustness.

## D  LIMITATIONS AND FUTURE WORK

In this work, we reconsider the significance of model complexity on prediction results, even seemingly straightforward models can yield competitive results. However, for high-dimensional and large-scale datasets, we need to further stack our proposed basic units to increase the model's capacity, enabling it to better adapt to intricate tasks, enhancing representational power, depth, feature extraction, and generalization performance. In practical scenarios, a substantial amount of time series data frequently incorporates spatial information. In future work, we intend to construct additional spatial modules built upon the LTSF-Conv basic unit to strengthen the adaptability of the model. Furthermore, we plan to extend LTSF-Conv to various other downstream tasks, such as classification and anomaly detection.

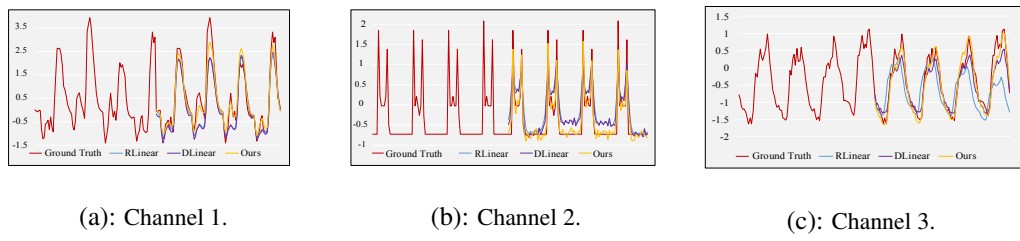

(a): Channel 1.   (b): Channel 2.   (c): Channel 3.

Figure 6: Forecasting results on Electricity with three random channels of different periods under the predict-96 setting.

---

[1]https://github.com/laiguokun/multivariate-time-series-data/

Table 13: Model efficiency comparison under $sl = 336$ on Electricity.

| Models | Series Length | Trainable Parameter (M) | GPU Memory (MiB) | Running Time (s / iter) | Iters (nums) | Training Time (s) | Inference Time (s) | GPUs (nums) |
|---|---|---|---|---|---|---|---|---|
| Conv | 96 | 0.0446 | 90.71 | 10.56 | 20 | 211.27 | 10.75 | 1 |
| | 192 | 0.0769 | 117.45 | 11.89 | 22 | 261.59 | 14.40 | 1 |
| | 336 | 0.1254 | 170.00 | 14.04 | 19 | 266.75 | 21.58 | 1 |
| | 720 | 0.2548 | 285.68 | 19.43 | 30 | 582.79 | 29.69 | 1 |
| | Avg | 0.1254 | 165.96 | 13.98 | 22.75 | 330.60 | 19.11 | 1 |
| DConv | 96 | 0.0763 | 119.99 | 11.68 | 26 | 303.65 | 15.40 | 1 |
| | 192 | 0.1410 | 146.42 | 13.08 | 29 | 379.41 | 24.64 | 1 |
| | 336 | 0.2380 | 207.08 | 15.08 | 19 | 286.46 | 34.72 | 1 |
| | 720 | 0.4968 | 349.91 | 20.06 | 20 | 401.24 | 59.55 | 1 |
| | Avg | 0.2380 | 205.85 | 14.98 | 23.5 | 342.69 | 33.58 | 1 |
| PatchTST | 96 | 0.9212 | 15978.00 | 265.30 | 50 | 13264.98 | 40.55 | 2 |
| | 192 | 1.4374 | 15999.82 | 265.87 | 19 | 5051.50 | 48.58 | 2 |
| | 336 | 2.2117 | 16034.15 | 266.07 | 49 | 13037.56 | 57.42 | 2 |
| | 720 | 4.2764 | 16116.24 | 265.34 | 22 | 5837.58 | 80.39 | 2 |
| | Avg | 2.2116 | 16032.05 | 265.65 | 35 | 9297.90 | 56.74 | 2 |
| Autoformer | 96 | 12.1439 | 4538.82 | 136.52 | 6 | 819.10 | 30.14 | 1 |
| | 192 | 12.1439 | 5223.64 | 156.88 | 7 | 1098.13 | 37.52 | 1 |
| | 336 | 12.1439 | 5513.15 | 185.88 | 6 | 1115.30 | 51.26 | 1 |
| | 720 | 12.1439 | 8529.48 | 273.23 | 7 | 1912.64 | 79.31 | 1 |
| | Avg | 12.1439 | 5951.27 | 188.18 | 6.5 | 1236.29 | 49.56 | 1 |
| Informer | 96 | 12.4537 | 2315.98 | 86.87 | 8 | 694.94 | 19.08 | 1 |
| | 192 | 12.4537 | 2444.42 | 96.50 | 15 | 1447.57 | 25.46 | 1 |
| | 336 | 12.4537 | 2086.85 | 113.19 | 8 | 905.48 | 33.97 | 1 |
| | 720 | 12.4537 | 2590.08 | 149.48 | 18 | 2690.69 | 56.90 | 1 |
| | Avg | 12.4537 | 2359.33 | 111.51 | 12.25 | 1434.67 | 33.85 | 1 |
| Reformer | 96 | 6.4382 | 2858.95 | 115.18 | 10 | 1151.81 | 22.61 | 1 |
| | 192 | 6.4382 | 3467.09 | 137.28 | 7 | 960.94 | 29.53 | 1 |
| | 336 | 6.4382 | 4378.95 | 170.68 | 10 | 1706.78 | 41.98 | 1 |
| | 720 | 6.4382 | 7049.06 | 263.51 | 13 | 3425.62 | 67.07 | 1 |
| | Avg | 6.4382 | 4438.52 | 171.66 | 10 | 1811.29 | 40.30 | 1 |
| LogTrans | 96 | 11.6657 | 3087.83 | 88.51 | 8 | 708.07 | 17.79 | 1 |
| | 192 | 11.6657 | 3375.00 | 98.27 | 14 | 1375.80 | 23.89 | 1 |
| | 336 | 11.6657 | 2996.17 | 113.14 | 7 | 791.99 | 34.02 | 1 |
| | 720 | 11.6657 | 5033.59 | 175.74 | 28 | 4920.77 | 56.55 | 1 |
| | Avg | 11.6657 | 3623.15 | 118.92 | 14.25 | 1949.16 | 33.07 | 1 |

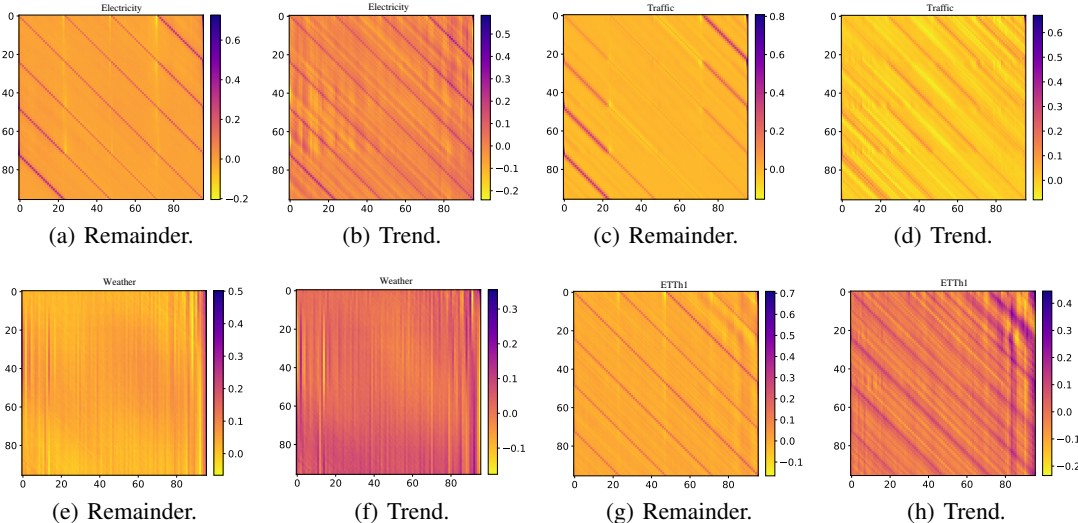

(a) Remainder.    (b) Trend.    (c) Remainder.    (d) Trend.

(e) Remainder.    (f) Trend.    (g) Remainder.    (h) Trend.

Figure 7: Visualization of the weights of DConv on different benchmarks. The X-axis represents the look-back window size, and the Y-axis represents the prediction length.

Table 14: Model efficiency comparison under $sl = 336$ on Weather.

| Models | Series Length | Trainable Parameter (M) | GPU Memory (MiB) | Running Time (s / iter) | Iters (nums) | Training Time (s) | Inference Time (s) | GPUs (nums) |
|---|---|---|---|---|---|---|---|---|
| Conv | 96 | 0.0332 | 6.19 | 7.58 | 11 | 83.42 | 5.91 | 1 |
| | 192 | 0.0655 | 8.40 | 7.54 | 20 | 150.73 | 6.51 | 1 |
| | 336 | 0.1140 | 12.09 | 7.97 | 15 | 119.48 | 7.58 | 1 |
| | 720 | 0.2434 | 21.94 | 8.62 | 21 | 181.04 | 8.82 | 1 |
| | Avg | 0.1140 | 12.15 | 7.93 | 16.75 | 133.67 | 7.20 | 1 |
| DConv | 96 | 0.0654 | 8.49 | 8.85 | 8 | 70.78 | 7.68 | 1 |
| | 192 | 0.1302 | 11.04 | 9.24 | 10 | 92.38 | 8.55 | 1 |
| | 336 | 0.2272 | 15.43 | 8.94 | 25 | 223.56 | 9.47 | 1 |
| | 720 | 0.4860 | 28.12 | 9.36 | 10 | 93.57 | 13.47 | 1 |
| | Avg | 0.2272 | 15.77 | 9.10 | 13.25 | 120.08 | 9.79 | 1 |
| PatchTST | 96 | 0.9212 | 1065.29 | 81.14 | 23 | 1866.22 | 13.29 | 1 |
| | 192 | 1.4374 | 1073.08 | 81.33 | 13 | 1057.25 | 14.44 | 1 |
| | 336 | 2.2117 | 1086.64 | 81.73 | 11 | 898.98 | 15.63 | 1 |
| | 720 | 4.2764 | 1122.29 | 82.86 | 12 | 994.39 | 19.41 | 1 |
| | Avg | 2.2116 | 1086.82 | 81.76 | 14.75 | 1204.20 | 15.69 | 1 |
| Autoformer | 96 | 10.6076 | 4465.52 | 266.57 | 6 | 1599.39 | 42.04 | 1 |
| | 192 | 10.6076 | 5142.19 | 303.50 | 6 | 1820.98 | 48.28 | 1 |
| | 336 | 10.6076 | 5125.08 | 363.88 | 6 | 2183.29 | 69.68 | 1 |
| | 720 | 10.6076 | 8016.37 | 532.23 | 6 | 3193.39 | 96.93 | 1 |
| | Avg | 10.6076 | 5687.29 | 366.54 | 6 | 2199.26 | 64.23 | 1 |
| Informer | 96 | 11.3782 | 2251.79 | 158.98 | 7 | 1112.86 | 19.923 | 1 |
| | 192 | 11.3782 | 2369.26 | 176.47 | 7 | 1235.31 | 22.22 | 1 |
| | 336 | 11.3782 | 1997.09 | 203.78 | 6 | 1222.69 | 32.52 | 1 |
| | 720 | 11.3782 | 2435.73 | 278.99 | 11 | 3068.91 | 45.62 | 1 |
| | Avg | 11.3782 | 2263.47 | 204.56 | 7.75 | 1659.94 | 30.07 | 1 |
| Reformer | 96 | 5.8235 | 2784.63 | 221.54 | 11 | 2436.91 | 27.35 | 1 |
| | 192 | 5.8235 | 3371.91 | 268.70 | 8 | 2149.58 | 32.99 | 1 |
| | 336 | 5.8235 | 4256.18 | 343.30 | 7 | 2403.12 | 46.07 | 1 |
| | 720 | 5.8235 | 6892.63 | 526.34 | 10 | 5263.38 | 72.81 | 1 |
| | Avg | 5.8235 | 4326.33 | 339.97 | 9 | 3063.25 | 44.80 | 1 |
| LogTrans | 96 | 10.5902 | 3018.15 | 168.47 | 6 | 1010.84 | 19.03 | 1 |
| | 192 | 10.5902 | 3297.23 | 189.21 | 6 | 1135.27 | 25.35 | 1 |
| | 336 | 10.5902 | 2892.28 | 220.57 | 6 | 1323.41 | 30.14 | 1 |
| | 720 | 10.5902 | 4877.56 | 344.36 | 6 | 2066.19 | 43.56 | 1 |
| | Avg | 10.5902 | 3521.30 | 230.65 | 6 | 1383.92 | 29.52 | 1 |

Table 15: Model efficiency comparison under $sl = 336$ on ETTm2.

| Models | Series Length | Trainable Parameter (M) | GPU Memory (MiB) | Running Time (s / iter) | Iters (nums) | Training Time (s) | Inference Time (s) | GPUs (nums) |
|---|---|---|---|---|---|---|---|---|
| Conv | 96 | 0.0326 | 2.40 | 6.11 | 26 | 158.79 | 5.84 | 1 |
| | 192 | 0.0649 | 3.46 | 6.18 | 18 | 111.20 | 5.92 | 1 |
| | 336 | 0.1135 | 5.19 | 6.27 | 27 | 169.33 | 6.32 | 1 |
| | 720 | 0.2429 | 9.79 | 6.80 | 16 | 108.74 | 7.15 | 1 |
| | Avg | 0.1134 | 5.21 | 6.34 | 21.75 | 137.02 | 6.31 | 1 |
| DConv | 96 | 0.0650 | 3.55 | 10.90 | 22 | 239.78 | 8.36 | 1 |
| | 192 | 0.1297 | 4.88 | 12.19 | 15 | 182.87 | 10.29 | 1 |
| | 336 | 0.2267 | 7.05 | 12.16 | 16 | 194.59 | 8.73 | 1 |
| | 720 | 0.4855 | 13.18 | 10.36 | 16 | 165.82 | 10.13 | 1 |
| | Avg | 0.2267 | 7.17 | 11.40 | 17.25 | 195.76 | 9.38 | 1 |
| PatchTST | 96 | 0.9212 | 365.60 | 29.28 | 11 | 322.07 | 8.27 | 1 |
| | 192 | 1.4374 | 376.49 | 30.79 | 11 | 338.71 | 8.91 | 1 |
| | 336 | 2.2117 | 387.91 | 30.04 | 10 | 300.44 | 9.56 | 1 |
| | 720 | 4.2764 | 421.48 | 30.20 | 9 | 271.81 | 10.37 | 1 |
| | Avg | 2.2116 | 387.87 | 30.08 | 10.25 | 308.26 | 9.28 | 1 |
| Autoformer | 96 | 10.5370 | 3046.10 | 191.56 | 9 | 1724.04 | 37.30 | 1 |
| | 192 | 10.5370 | 3378.39 | 215.69 | 7 | 1509.83 | 41.80 | 1 |
| | 336 | 10.5370 | 2909.60 | 246.71 | 10 | 2467.10 | 53.38 | 1 |
| | 720 | 10.5370 | 4449.39 | 346.27 | 6 | 2077.63 | 72.89 | 1 |
| | Avg | 10.5370 | 3445.87 | 250.06 | 8 | 1944.65 | 51.35 | 1 |
| Informer | 96 | 11.3290 | 2258.83 | 161.44 | 6 | 968.66 | 26.85 | 1 |
| | 192 | 11.3290 | 2381.14 | 180.83 | 6 | 1084.99 | 30.28 | 1 |
| | 336 | 11.3290 | 2005.51 | 204.47 | 8 | 1635.76 | 36.36 | 1 |
| | 720 | 11.3290 | 3128.20 | 282.11 | 15 | 4231.67 | 50.62 | 1 |
| | Avg | 11.3290 | 2443.42 | 207.21 | 8.75 | 1980.27 | 36.03 | 1 |
| Reformer | 96 | 5.7953 | 2780.49 | 208.82 | 6 | 1252.94 | 31.75 | 1 |
| | 192 | 5.7953 | 3368.69 | 253.63 | 6 | 1521.75 | 37.71 | 1 |
| | 336 | 5.7953 | 4249.20 | 317.01 | 8 | 2536.10 | 47.34 | 1 |
| | 720 | 5.7953 | 6884.70 | 489.97 | 7 | 3429.76 | 73.77 | 1 |
| | Avg | 5.7953 | 4320.77 | 317.36 | 6.75 | 2185.14 | 47.64 | 1 |
| LogTrans | 96 | 10.5411 | 3015.32 | 159.83 | 9 | 1438.46 | 22.07 | 1 |
| | 192 | 10.5411 | 3293.75 | 178.98 | 7 | 1252.85 | 25.12 | 1 |
| | 336 | 10.5411 | 2888.15 | 207.05 | 7 | 1449.32 | 28.89 | 1 |
| | 720 | 10.5411 | 4870.88 | 322.14 | 7 | 2254.95 | 42.60 | 1 |
| | Avg | 10.5411 | 3517.03 | 217.00 | 7.5 | 1598.89 | 29.67 | 1 |

Table 16: The performance comparison of two structures. $Ori$ represents the original *encoder-decoder* structure, and a modified version is denoted as $Half\text{-}Ex$, which retains half of the original design, only the decoder part.

| Methods | | FEDformer | | | | Autoformer | | | | LogTrans | | | |
|---|---|---|---|---|---|---|---|---|---|---|---|---|---|
| | | MSE | | MAE | | MSE | | MAE | | MSE | | MAE | |
| Predict Length | | Ori | Half-Ex | Ori | Half-Ex | Ori | Half-Ex | Ori | Half-Ex | Ori | Half-Ex | Ori | Half-Ex |
| ETTh1 | 96 | 0.376 | 0.374 | 0.419 | 0.414 | 0.449 | 0.387 | 0.459 | 0.423 | 0.878 | 0.839 | 0.74 | 0.708 |
| | 192 | 0.42 | 0.422 | 0.448 | 0.446 | 0.5 | 0.437 | 0.482 | 0.447 | 1.037 | 1.056 | 0.824 | 0.823 |
| | 336 | 0.459 | 0.447 | 0.465 | 0.463 | 0.521 | 0.502 | 0.496 | 0.491 | 1.238 | 1.005 | 0.932 | 0.811 |
| | 720 | 0.506 | 0.487 | 0.507 | 0.492 | 0.514 | 0.493 | 0.512 | 0.501 | 1.135 | 1.212 | 0.852 | 0.914 |
| | Avg | 0.44 | **0.432** | 0.459 | **0.453** | 0.496 | **0.454** | 0.487 | **0.465** | 1.072 | **1.028** | 0.837 | **0.814** |
| ETTh2 | 96 | 0.346 | 0.34 | 0.388 | 0.383 | 0.358 | 0.405 | 0.397 | 0.422 | 2.116 | 1.772 | 1.197 | 1.074 |
| | 192 | 0.429 | 0.431 | 0.439 | 0.438 | 0.456 | 0.426 | 0.452 | 0.433 | 4.315 | 3.986 | 1.635 | 1.601 |
| | 336 | 0.496 | 0.481 | 0.487 | 0.479 | 0.482 | 0.452 | 0.486 | 0.463 | 4.511 | 3.722 | 1.758 | 1.588 |
| | 720 | 0.463 | 0.466 | 0.474 | 0.478 | 0.515 | 0.467 | 0.511 | 0.482 | 3.188 | 3.161 | 1.54 | 1.473 |
| | Avg | 0.433 | **0.429** | 0.447 | **0.444** | 0.452 | **0.437** | 0.461 | **0.45** | 3.532 | **3.16** | 1.532 | **1.434** |
| ETTm1 | 96 | 0.379 | 0.361 | 0.419 | 0.408 | 0.505 | 0.517 | 0.475 | 0.484 | 0.6 | 0.548 | 0.546 | 0.529 |
| | 192 | 0.426 | 0.404 | 0.441 | 0.432 | 0.553 | 0.557 | 0.496 | 0.504 | 0.837 | 0.659 | 0.7 | 0.602 |
| | 336 | 0.445 | 0.454 | 0.459 | 0.461 | 0.621 | 0.541 | 0.537 | 0.496 | 1.124 | 1.011 | 0.832 | 0.77 |
| | 720 | 0.543 | 0.508 | 0.49 | 0.488 | 0.671 | 0.559 | 0.561 | 0.507 | 1.153 | 1.079 | 0.82 | 0.806 |
| | Avg | 0.448 | **0.431** | 0.452 | **0.447** | 0.587 | **0.543** | 0.517 | **0.497** | 0.928 | **0.824** | 0.724 | **0.676** |
| ETTm2 | 96 | 0.203 | 0.188 | 0.287 | 0.281 | 0.255 | 0.214 | 0.339 | 0.296 | 0.768 | 0.555 | 0.642 | 0.547 |
| | 192 | 0.269 | 0.255 | 0.328 | 0.322 | 0.281 | 0.273 | 0.34 | 0.33 | 0.989 | 0.854 | 0.757 | 0.713 |
| | 336 | 0.325 | 0.318 | 0.366 | 0.363 | 0.339 | 0.328 | 0.372 | 0.363 | 1.334 | 1.153 | 0.872 | 0.794 |
| | 720 | 0.421 | 0.427 | 0.415 | 0.422 | 0.422 | 0.428 | 0.419 | 0.421 | 3.048 | 3.048 | 1.328 | 1.31 |
| | Avg | 0.304 | **0.297** | 0.349 | **0.347** | 0.324 | **0.31** | 0.367 | **0.352** | 1.534 | **1.402** | 0.899 | **0.841** |
| Traffic | 96 | 0.587 | 0.569 | 0.366 | 0.352 | 0.613 | 0.615 | 0.388 | 0.396 | 0.684 | 0.676 | 0.384 | 0.369 |
| | 192 | 0.604 | 0.599 | 0.373 | 0.373 | 0.616 | 0.626 | 0.382 | 0.394 | 0.685 | 0.674 | 0.39 | 0.367 |
| | 336 | 0.621 | 0.614 | 0.383 | 0.376 | 0.622 | 0.619 | 0.337 | 0.385 | 0.734 | 0.658 | 0.408 | 0.358 |
| | 720 | 0.626 | 0.624 | 0.382 | 0.38 | 0.66 | 0.635 | 0.408 | 0.395 | 0.717 | 0.679 | 0.396 | 0.363 |
| | Avg | 0.609 | **0.601** | 0.376 | **0.37** | 0.627 | **0.623** | 0.378 | **0.378** | 0.705 | **0.671** | 0.394 | **0.364** |
| Exchange | 96 | 0.148 | 0.135 | 0.278 | 0.263 | 0.197 | 0.153 | 0.323 | 0.285 | 0.968 | 0.59 | 0.812 | 0.594 |
| | 192 | 0.271 | 0.278 | 0.38 | 0.384 | 0.3 | 0.268 | 0.369 | 0.378 | 1.04 | 1.076 | 0.851 | 0.783 |
| | 336 | 0.46 | 0.448 | 0.5 | 0.492 | 0.509 | 0.45 | 0.524 | 0.499 | 1.659 | 1.28 | 1.081 | 0.923 |
| | 720 | 1.195 | 1.163 | 0.841 | 0.824 | 1.447 | 1.141 | 0.941 | 0.827 | 1.941 | 2.545 | 1.127 | 1.366 |
| | Avg | 0.518 | **0.506** | 0.499 | **0.49** | 0.613 | **0.503** | 0.539 | **0.497** | 1.402 | **1.372** | 0.967 | **0.916** |
| ILI | 24 | 3.228 | 3.171 | 1.26 | 1.235 | 3.483 | 3.277 | 1.287 | 1.255 | 4.48 | 4.226 | 1.444 | 1.325 |
| | 36 | 2.679 | 2.598 | 1.08 | 1.056 | 3.103 | 2.772 | 1.148 | 1.118 | 4.799 | 4.598 | 1.467 | 1.405 |
| | 48 | 2.622 | 2.48 | 1.078 | 1.045 | 2.669 | 2.697 | 1.085 | 1.114 | 4.8 | 4.967 | 1.468 | 1.462 |
| | 60 | 2.857 | 2.721 | 1.157 | 1.118 | 2.77 | 2.737 | 1.125 | 1.122 | 5.278 | 4.772 | 1.56 | 1.461 |
| | Avg | 2.846 | **2.742** | 1.143 | **1.113** | 3 | **2.87** | 1.161 | **1.152** | 4.839 | **4.64** | 1.484 | **1.413** |
| Electricity | 96 | 0.193 | 0.184 | 0.308 | 0.299 | 0.201 | 0.195 | 0.317 | 0.307 | 0.258 | 0.271 | 0.357 | 0.365 |
| | 192 | 0.201 | 0.196 | 0.315 | 0.31 | 0.222 | 0.225 | 0.334 | 0.335 | 0.266 | 0.278 | 0.368 | 0.37 |
| | 336 | 0.214 | 0.212 | 0.329 | 0.326 | 0.231 | 0.251 | 0.338 | 0.359 | 0.28 | 0.283 | 0.38 | 0.373 |
| | 720 | 0.246 | 0.239 | 0.355 | 0.348 | 0.254 | 0.28 | 0.361 | 0.383 | 0.283 | 0.289 | 0.376 | 0.375 |
| | Avg | 0.213 | **0.207** | 0.326 | **0.32** | **0.227** | 0.237 | **0.337** | 0.346 | **0.271** | 0.28 | **0.37** | **0.37** |
| Weather | 96 | 0.217 | 0.264 | 0.239 | 0.345 | 0.266 | 0.225 | 0.336 | 0.3 | 0.458 | 0.435 | 0.49 | 0.462 |
| | 192 | 0.276 | 0.254 | 0.336 | 0.316 | 0.307 | 0.298 | 0.367 | 0.353 | 0.658 | 0.459 | 0.589 | 0.474 |
| | 336 | 0.339 | 0.359 | 0.38 | 0.395 | 0.359 | 0.345 | 0.395 | 0.382 | 0.797 | 0.501 | 0.652 | 0.504 |
| | 720 | 0.403 | 0.398 | 0.428 | 0.411 | 0.419 | 0.411 | 0.428 | 0.42 | 0.675 | 0.668 | 1.13 | 0.587 |
| | Avg | **0.308** | 0.318 | **0.345** | 0.366 | 0.337 | **0.319** | 0.381 | **0.363** | 0.647 | **0.515** | 0.715 | **0.506** |

Table 17: Comparison of practical efficiency of LTSF-Linear and LTSF-Conv under $sl = 336$ on the Electricity. All results are the average test of three runs.

| Models | | RLinear | | DLinear | | Conv |
|---|---|---|---|---|---|---|
| | | CI | S | CI | S | - |
| Trainable Parameter (M) | 96 | 10.39 | 0.03 | 31.15 | 0.1 | 0.04 |
| | 192 | 20.77 | 0.07 | 62.31 | 0.19 | 0.08 |
| | 336 | 36.35 | 0.11 | 109.04 | 0.34 | 0.13 |
| | 720 | 77.89 | 0.24 | 233.66 | 0.73 | 0.25 |
| GPU Memory (MiB) | 96 | 536.93 | 307.47 | 716.18 | 398.01 | 336.79 |
| | 192 | 737.83 | 400.058 | 1190.77 | 501.78 | 452.89 |
| | 336 | 1148.48 | 581.36 | 1905.77 | 661.24 | 634.2 |
| | 720 | 2263.83 | 1064.83 | 3974.65 | 1148.93 | 1117.67 |
| Running Time (s / epoch) | 96 | 68.47 | 7.83 | 50.81 | 8.48 | 7.18 |
| | 192 | 73.13 | 9.095 | 53.46 | 9.96 | 8.47 |
| | 336 | 78.92 | 11.19 | 64.29 | 13.96 | 12.99 |
| | 720 | 94.57 | 16.25 | 93.61 | 20.4 | 16.54 |

