# OpenReview forum: "The Power of Minimalism in Long Sequence Time-series Forecasting"
_ICLR.cc/2024/Conference — Submitted to ICLR 2024_

### Official Review · Reviewer_gotF · 2023-10-29

**Soundness:** 3 good
**Presentation:** 4 excellent
**Contribution:** 3 good
**Rating:** 6
**Confidence:** 5

**Summary:**

The paper studies the problem of time series forecasting. The paper investigates the performance of a simple model, namely a one-layer convolutional network applied to every feature independently and combined with a linear layer. The paper shows that such a simple approach could improve upon the existing baselines in most of the cases with significantly reduced computational costs.

**Strengths:**

1. The paper shows that a simple network structure could be very effective in the widely used time-series forecasting datasets. The paper thus provides a valuable baseline that future methods should all consider when dealing with such kinds of tasks.

2. The paper conducts extensive experiments and studies to make their results convincing.

**Weaknesses:**

1. As also mentioned in the paper, time series with multiple periodic intervals could not be captured by a single convolutional layer. I think it might make the paper stronger by generating synthetic data with various periodic behaviors and testing various models on it.

2. Every univariate is now processed independently in the current convolutional network. Is there a specific reason for doing so except for the efficiency concerns? How would the performance change if we also include the feature dimension in the convolutional filter?

3. Why just one layer of convolution? How does the performance change if multiple layers are applied? This may help to capture longer periodic patterns or even help with the multi-period issue.

4. As also mentioned in the paper, the effectiveness of relatively simple models such as DLinear and the convolutional network may largely depend on the nature of the current tasks. For much more complex time series with more features and periodic complexities, such simpler methods may not be as good as the transformer-based models.

5. How do we determine the kernel size of the convolution, which should be critical for the forecasting task, especially for the cases where we don't know the periodic interval of the data streams?

**Questions:**

Please check the weaknesses part.

Update after the rebuttal:
Thanks for the detailed response. It addresses some of my concerns but some remain. E.g., there is no empirical evidence to support the claim of estimating the period. And the solution for more complex tasks is reasonable but not convincing enough. Overall, I think the work could serve as a solid baseline for the time series forecasting tasks, so I keep my score for weak acceptance.

---

> ### Author Response · Authors · 2023-11-18
> **Response to Reviewer gotF**
>
> We sincerely appreciate your detailed comments and positive ranking. We provide point-wise responses to your concerns below.
>
> Q1 : *Every univariate is now processed independently in the current convolutional network. Is there a specific reason for doing so except for the efficiency concerns? How would the performance change if we also include the feature dimension in the convolutional filter?*
> * Many thanks for your careful reading and suggestion. In LTSF tasks, maintaining channel independence has been observed to improve prediction performance as compared to channel mixing, as reported in prior research [2]. **Our model utilizes group-wise convolution to cleverly achieve channel independence, while concurrently reducing model complexity.** However, standard CNN uses the idea of channel mixing, which suffers from noise disturbance among the channels and reduces performance. We have conducted additional ablation experiments about depth-wise CNN and general CNN. The results are in Section C.2, Page 13 of the original manuscript. Channel independence helps reduce information confusion. When each channel focuses on capturing specific time series long-term patterns, the model is more capable of distinguishing and understanding these patterns without the noise that channel mixing might introduce. If we also include the feature dimension in the convolutional filter, the performance will degrade.
> * Moreover, we also apply channel independence and channel dependence to the MLPs-based models respectively. The results can be found in Table 5 of the attached PDF file. Overall, the accuracy of the CI-based models was higher than the CD-based models.
>
> Q2 : *Why just one layer of convolution? How does the performance change if multiple layers are applied? This may help to capture longer periodic patterns or even help with the multi-period issue.*
> * We are grateful for your constructive comments. **We want to emphasize that this manuscript does not only aim to showcase the contributions of SOTA results. Simple yet valuable basic units can serve as foundational building blocks, laying the groundwork for scalable and complex networks.**  We firmly believe that LTSF-Conv models serve as straightforward yet competitive basic units, exhibiting great potential for further expansion of complex structures. As the reviewers pointed out, based on our extended experiments, when applying complex network structures, the performance will be further improved. In the future, there is the potential for more valuable research to be explored in this new track.
>
> Q3: *Also mentioned in the paper, the effectiveness of relatively simple models such as DLinear and the convolutional network may largely depend on the nature of the current tasks. For much more complex time series with more features and periodic complexities, such simpler methods may not be as good as the transformer-based models.*
> * Thank you for pointing out this. When dealing with more complex datasets, increasing the depth, or introducing additional hierarchical structures to the network helps the model adapt better to the diversity of the data. For LTSF-Conv basic unit, exhibiting great potential for further expansion of complex structures. Whether based on MLPs, simpler convolution, or transformers, each model has contributed to the field of long sequence prediction. The development of these models has advanced the field of LTSF, providing diverse modeling choices for various tasks.
>
> Q4: *How do we determine the kernel size of the convolution, which should be critical for the forecasting task, especially for the cases where we don't know the periodic interval of the data streams?*
> * Thank you for pointing out this. In essence, finding the most suitable convolutional kernel size is a dynamic process, combining experimentation and consideration of specific data intricacies for optimal model performance. It's important to iterate and evaluate the model's performance to find the optimal kernel size for your specific forecasting task. In the absence of known periodic intervals, we usually employ a multi-scale analysis strategy, which enables the model to capture features at different time scales without relying on prior knowledge of periodicity. We can also use cross-validation to assess the performance of different kernel sizes.

---

> > ### Author Response · Authors · 2023-11-21
> > **Response to Reviewer gotF**
> >
> > Q5: *As also mentioned in the paper, time series with multiple periodic intervals could not be captured by a single convolutional layer. I think it might make the paper stronger by generating synthetic data with various periodic behaviors and testing various models on it. The paper conducts extensive experiments and studies to make their results convincing*.
> > * Many thanks for your careful reading and suggestion. We are sorry for the late submission due to the addition of two new datasets and some new baselines. We supplemented additional experiments to further verify the validity of LTSF-Conv models in LTSF tasks.
> >
> > * Solar power prediction is a pivotal concern within the realm of renewable energy, wielding significant influence across diverse domains. We add two supplementary datasets from real-world applications, namely the Solar-Jinta and Solar-Alabama benchmarks. We think public datasets are more convincing. Meanwhile, we also add several superior baselines on Solar-Energy datasets. Solar-Jinta records seven key meteorological factors of solar radiation, collected by the hour. Solar-Alabama contains the solar power production of 137 PV plants in the USA, with a data granularity of 10 minutes. The solar power of different PV plants is influenced by varying geographical and weather conditions. You can download the datasets at https://github.com/laiguokun/multivariate-time-series-data. Table 7 provides a summary of prediction results from several baselines on the Solar-Energy datasets. It can be observed that DConv outperforms the other baselines for most horizons by a large margin. Moreover, compared to other datasets, the Solar-Jinta benchmark has a smaller size (num 8761). The experimental results also indicate that our model performs equally well on small datasets, ensuring its generalizability and robustness.  We firmly believe that LTSF-Conv models as competitive basic units, exhibiting great potential for further expansion of complex network structures.
> >
> >
> >
> >
> >
> > |  |  |  |  | MSE |  |  |  |  | MAE |  |  |
> > |---|:---:|:---:|:---:|:---:|:---:|:---:|:---:|:---:|:---:|:---:|:---:|
> > | Benchmarks | Methods | 96 | 192 | 336 | 720 | Avg | 96 | 192 | 336 | 720 | Avg |
> > | Solar-Jinta | DConv | **0.480** | **0.521** | **0.539** | **0.648** | **0.547** | 0.481 | **0.496** | **0.522** | **0.573** | **0.518** |
> > |  | PatchTST | 0.491  | 0.602  | 0.617  | 0.710  | 0.605  | **0.464** | 0.523  | 0.533  | 0.601  | 0.530  |
> > |  | Flowformer | 0.646  | 0.792  | 0.748  | 1.065  | 0.812  | 0.565  | 0.624  | 0.611  | 0.783  | 0.645  |
> > |  | Reformer | 0.954  | 0.934  | 0.993  | 0.995  | 0.969  | 0.773  | 0.757  | 0.742  | 0.730  | 0.751  |
> > |  | Informer | 0.718  | 0.821  | 0.847  | 1.102  | 0.872  | 0.616  | 0.656  | 0.737  | 0.855  | 0.716  |
> > |  | LogTrans | 0.704  | 0.817  | 0.754  | 1.012  | 0.821  | 0.594  | 0.651  | 0.642  | 0.762  | 0.662  |
> > |  | DLinear | 0.523  | 0.588  | 0.638  | 0.734  | 0.620  | 0.496  | 0.531  | 0.562  | 0.616  | 0.551  |
> > | Solar-Energy | DConv | **0.173** | **0.189** | **0.201** | **0.206** | **0.192** | **0.228** | **0.237** | **0.244** | **0.245** | **0.239** |
> > |  | PatchTST | 0.178  | 0.192  | 0.203  | 0.218  | 0.197  | 0.248  | 0.265  | 0.267  | 0.274  | 0.263  |
> > |  | Flowformer | 0.190  | 0.267  | 0.279  | 0.243  | 0.245  | 0.229  | 0.263  | 0.268  | 0.252  | 0.253  |
> > |  | Reformer | 0.195  | 0.223  | 0.253  | 0.281  | 0.238  | 0.234  | 0.252  | 0.286  | 0.286  | 0.265  |
> > |  | Informer | 0.195  | 0.220  | 0.259  | 0.246  | 0.230  | 0.241  | 0.241  | 0.274  | 0.266  | 0.256  |
> > |  | LogTrans | 0.219  | 0.217  | 0.224  | 0.241  | 0.225  | 0.240  | 0.244  | 0.258  | 0.269  | 0.252  |
> > |  | DLinear | 0.221  | 0.231  | 0.247  | 0.255  | 0.239  | 0.294  | 0.301  | 0.317  | 0.314  | 0.307  |
> >
> > **Table 7**: Multivariate long-term time series forecasting results on Solar benchmarks.
> >
> > **We look forward to your further feedback**.

---

### Official Review · Reviewer_1dcB · 2023-10-31

**Soundness:** 2 fair
**Presentation:** 2 fair
**Contribution:** 2 fair
**Rating:** 3
**Confidence:** 4

**Summary:**

This paper aims to address the challenge of long-term time series forecasting (LTSF) and introduces LTSF-Conv models as a solution. The authors discuss the limitations of existing methods, such as Transformer-based models and MLP-based models, and highlight the need for a balance between performance and efficiency. The experiments show that the proposed LTSF-Conv models, based on convolutional neural networks (CNNs), consistently outperform complex Transformer-based models and state-of-the-art MLP-based models, while maintaining efficiency. The paper provides some insights into input window sizes, encoder-decoder structures, and handling time series with multiple periods among channels.

**Strengths:**

This paper has the following advantages:

1. Novel solution: The paper introduces the LTSF-Conv model as a new approach to address long-term time series forecasting. By utilizing convolutional neural networks (CNNs), the model outperforms complex Transformer-based and MLP-based models in most cases while maintaining efficiency.

2. Empirical research and extensive experiments: The authors conduct comprehensive experimental evaluations on multiple real-world datasets across various domains such as weather, traffic, and electricity. The results consistently demonstrate that LTSF-Conv models outperform other complex models in terms of average performance. The paper provides concrete performance comparison data to support their findings.

3. Analysis and discussion of existing models' limitations: The paper thoroughly analyzes the limitations of existing Transformer-based and MLP-based solutions, particularly in handling long-term time series and multi-channel data. This analysis helps to understand their constraints and guides future research.

4. Efficiency and reduced computational resources: Compared to complex models, LTSF-Conv models achieve high performance while significantly reducing computational resource requirements. This is particularly valuable in practical applications with limited computing resources, enhancing the model's practical usability and scalability.

5. Insights for other aspects in the field: The paper also explores issues related to input window sizes, encoder-decoder structures, and handling multi-channel time series, providing valuable insights for future research in the LTSF domain.

**Weaknesses:**

Based on your understanding of the AI industry, you believe this paper has the following shortcomings:

1. Lack of innovation:
   - The model used in the paper consists of only two layers of convolutional networks, along with a decomposition of trend and periodic components. The loss function used is the classical MSE loss. There is a lack of innovation in the model design.
   - The innovation mainly lies in explaining the good performance of the simple convolutional model. However, the paper only provides a simple "proof" that convolutional kernels larger than a certain duration can capture periodic information shorter than that duration. The subsequent heatmaps only qualitatively observe that the model captures some periodic information, without explaining why the simple convolutional model is competitive.
   - Obvious conclusions, such as the smaller memory footprint, shorter training and inference time, and fewer parameters of the simple model, are extensively analyzed and explained in the paper.

2. Experimental limitations:
   - The paper lacks several important baseline models based on CNN architectures, such as SCINet and TimesNet, in the Conv-based model category (this is particularly severe, as all models in this category are proposed in this paper).
   - MLP-based models lack models like N-Hits, and there is also a lack of references to the aforementioned models.
   - The discussion of "model performance with respect to lookback" in Section C.1 lacks the inclusion of DConv. Considering Table 1 and Table 2, which compare the model results, the "best" model used in the tables is actually the model with a lookback of 1600, which naturally performs better than other baseline models with smaller lookback values that have not reached their optimal states. Moreover, even the "best" model is outperformed by PatchTST, which does not have data with lookback values of 720, 1000, and 1600.
   - The performance of the proposed models in complex datasets like Traffic is poor.

3. Presentation issues:
   - The quality of the figures illustrating the model is low and overly simplified.
   - There are formatting issues with the caption of Figure 6.

In summary, the identified shortcomings of the paper include a lack of innovation in the model design, experimental limitations in terms of missing baseline models and dataset performance, and presentation issues with figures and captions.

**Questions:**

What I'm concerned about are listed in the weakness. I won't refuse to raise my points if the authors can address my concerns.

---

> ### Author Response · Authors · 2023-11-18
> **Response to Reviewer 1dcB**
>
> Many thanks for your careful reading and suggestion. We appreciate that you found us some advantages.
>
> Meanwhile, we would like to provide some insights into the contribution of this paper. When the DLinear model [1] was initially introduced, its simple linear structure sparked considerable controversy. However, it ultimately won the Best Paper Award, defying complex Transformer models in the LTSF field. Subsequently, more works have been dedicated to refining and advancing linear models, including architectural modifications and innovative training methodologies. **We want to emphasize that our manuscript does not only aim to showcase the contributions of SOTA results. Our model structure is as simple as DLinear, while extensive experiments have consistently demonstrated that depth-wise convolutional units possess inherent advantages over linear units (such as DLinear, and RLinear) in LTSF.** Linear mappings often struggle to capture intricate dependencies when handling data with multiple periods among channels. Although modeling each channel independently can alleviate this issue, it will significantly increase the computational cost. More experimental analyses are described in Section 7, Page 9.  **Based on our basic unit, more complex networks can be proposed to further improve the prediction performance of such tasks. In the future, there is the potential for more valuable research to be explored in this new track.**
>
> We would like to address your concerns point-by-point below:
>
> Q1 : *The paper lacks several important baseline models based on CNN architectures, such as SCINet and TimesNet, in the Conv-based model category (this is particularly severe, as all models in this category are proposed in this paper). MLP-based models lack models like N-Hits, and there is also a lack of references to the aforementioned models.*
> * Thank you for your comments. Due to space limitations, we compared representative SOTA models from recent years in original our manuscripts. Following the reviewer’s suggestion, **we have added SCINet, TimesNet, MICN, and N-Hits baseline models.** We adopt their official codes and only change the length of input sequences. The results are summarized in Table 2. In Table 2, we conducted an experiment within a broader window size range of \{96, 336, 512, 720, 1600\} for a fair comparison. **It can be found in the attached supplementary material file.**  The results demonstrate that LTSF-Conv models consistently surpass all SCINet, TimesNet, MICN, and N-Hits on seven LTSF benchmarks. The symbol * denotes the experimental results we conducted after increasing the input length (re-implementation). For N-Hits, authors have adopted an extended step size hyperparameter search, and we directly adopt the results. We will add the related references in the revised paper.
>
> Q2 : *Considering Table 1 and Table 2, which compare the model results, the "best" model used in the tables is actually the model with a lookback of 1600, which naturally performs better than other baseline models with smaller lookback values that have not reached their optimal states. Moreover, even the "best" model is outperformed by PatchTST, which does not have data with lookback values of 720, 1000, and 1600.*
> * Thank you for pointing out this.  Previous research [1] has also shown that Transformer-based baselines have not benefited from a longer look-back window. Their performance fluctuates or gets worse as the input lengths increase. We have conducted experiments to analyze it. **The reason why most transformer models do not increase the lookback window length is because the performance will become worse.** More analysis of the look-back windows is described in Section C.1, Page 12. in the original Appendix.
> * We have confidence in the fairness of our previous comparison. **The experimental results not only include Conv-Best and DConv-Best but also the default look-back window length is set at 512, such as Conv and Dconv.** This is using the same lookback window length as PatchTST.
> * Different from other Transformer-based baselines, PatchTST can extend the lookback window to 512, but does not benefit from longer windows. It's important to note that this extension comes at the cost of a substantial increase in the model’s computational resource requirements. On our server resources, PatchTST ran out of GPU memory for a look-back window size greater than 720.
> * To alleviate your concerns, we switched to a higher-performance server to validate PatchTST with lookback-window size $\in$ \{720, 1600\}. Since our model only experiments from the extended window {336, 512, 720, 1600}, we ignore the 1000 step size. **The results are in Table 3 of the attached supplementary material file.** As expected, **PatchTST did not benefit from longer lookback windows. As the input length increases, the significant memory overhead poses limitations for its practical application.**

---

> > ### Author Response · Authors · 2023-11-18
> > **Response to Reviewer 1dcB**
> >
> > Q3 : *The discussion of "model performance with respect to lookback" in Section C.1 lacks the inclusion of DConv.*
> > * Thank you for pointing out this. Due to the computational cost of Dconv and Conv is similar, we do not add it in Figure 5 of the original manuscript. The detailed efficiency comparison is described in Tables 6-8, Page 16-17. in the original Appendix.
> >
> > Q4 : *There are formatting issues with the caption of Figure 6. The quality of the figures illustrating the model is low and overly simplified.*
> > * Thank you so much for your careful check. We are sorry that we ignored the issues of font size on the vertical axis in Figure 6. We have revised it. Following the reviewer’s suggestion, we have made modifications to Section 4.1 of the original manuscript to provide a more detailed explanation of the model.
> >
> >
> > | Methods |  | Conv|  | DConv|  | PatchTST|  | Conv |  | DConv|  |PatchTST| |
> > |:---:|:---:|:---:|:---:|:---:|:---:|:---:|:---:|:---:|:---:|:---:|:---:|:---:|:---:|
> > | Series Length |  |  |  |720|  |  |  |  |  |1600|  |  |  |
> > | Metric |  | MSE | MAE | MSE | MAE | MSE | MAE | MSE | MAE | MSE | MAE | MSE | MAE |
> > | Weather | 96 | 0.141  | 0.189  | 0.167  | 0.220  | 0.156  | 0.208  | 0.141  | 0.196  | 0.166  | 0.222  | 0.159  | 0.214  |
> > |  | 192 | 0.183  | 0.231  | 0.212  | 0.259  | 0.197  | 0.245  | 0.183  | 0.239  | 0.209  | 0.260  | 0.202  | 0.253  |
> > |  | 336 | 0.233  | 0.272  | 0.257  | 0.294  | 0.247  | 0.284  | 0.232  | 0.277  | 0.253  | 0.293  | 0.309  | 0.350  |
> > |  | 720 | 0.303  | 0.324  | 0.318  | 0.339  | 0.311  | 0.333  | 0.296  | 0.324  | 0.307  | 0.334  | 0.306  | 0.334  |
> > |  | Avg | **0.215** | **0.254** | 0.239  | 0.278  | 0.228  | 0.268  | **0.213** | **0.259** | 0.234  | 0.277  | 0.244  | 0.288  |
> > | Electricity | 96 | 0.131  | 0.225  | 0.133  | 0.228  | 0.133  | 0.229  | 0.129  | 0.225  | 0.130  | 0.226  | 0.140  | 0.245  |
> > |  | 192 | 0.146  | 0.239  | 0.145  | 0.242  | 0.149  | 0.245  | 0.144  | 0.239  | 0.145  | 0.239  | 0.156  | 0.258  |
> > |  | 336 | 0.161  | 0.257  | 0.162  | 0.258  | 0.168  | 0.266  | 0.159  | 0.255  | 0.161  | 0.256  | 0.173  | 0.277  |
> > |  | 720 | 0.200  | 0.288  | 0.202  | 0.291  | 0.206  | 0.298  | 0.195  | 0.286  | 0.199  | 0.288  | 0.212  | 0.308  |
> > |  | Avg | **0.160** | **0.252** | 0.161  | 0.255  | 0.164  | 0.260  | **0.157** | **0.251** | 0.159  | 0.252  | 0.170  | 0.272  |
> > | ETTm2 | 96 | 0.161  | 0.251  | 0.162  | 0.252  | 0.166  | 0.259  | 0.161  | 0.257  | 0.161  | 0.255  | 0.170  | 0.265  |
> > |  | 192 | 0.218  | 0.291  | 0.216  | 0.290  | 0.221  | 0.297  | 0.213  | 0.296  | 0.214  | 0.292  | 0.226  | 0.305  |
> > |  | 336 | 0.272  | 0.329  | 0.268  | 0.326  | 0.272  | 0.331  | 0.258  | 0.329  | 0.259  | 0.325  | 0.279  | 0.341  |
> > |  | 720 | 0.351  | 0.387  | 0.347  | 0.378  | 0.350  | 0.380  | 0.325  | 0.378  | 0.325  | 0.369  | 0.344  | 0.379  |
> > |  | Avg | 0.251  | 0.315  | **0.248** | **0.312** | 0.252  | 0.317  | **0.239** | 0.315  | 0.240  | **0.310** | 0.255  | 0.323  |
> > | ETTm1 | 96 | 0.294  | 0.341  | 0.306  | 0.349  | 0.299  | 0.352  | 0.298  | 0.348  | 0.307  | 0.354  | 0.314  | 0.367  |
> > |  | 192 | 0.333  | 0.362  | 0.336  | 0.366  | 0.340  | 0.377  | 0.332  | 0.368  | 0.334  | 0.370  | 0.341  | 0.382  |
> > |  | 336 | 0.363  | 0.382  | 0.365  | 0.384  | 0.376  | 0.398  | 0.356  | 0.384  | 0.356  | 0.384  | 0.369  | 0.399  |
> > |  | 720 | 0.413  | 0.410  | 0.414  | 0.411  | 0.417  | 0.422  | 0.394  | 0.409  | 0.394  | 0.406  | 0.407  | 0.421  |
> > |  | Avg | **0.351** | **0.374** | 0.355  | 0.378  | 0.358  | 0.387  | **0.345** | **0.377** | 0.348  | 0.379  | 0.358  | 0.392  |
> > | ETTh1 | 96 | 0.377  | 0.404  | 0.375  | 0.398  | 0.379  | 0.411  | 0.391  | 0.416  | 0.387  | 0.410  | 0.380  | 0.413  |
> > |  | 192 | 0.417  | 0.428  | 0.413  | 0.421  | 0.416  | 0.433  | 0.425  | 0.437  | 0.422  | 0.432  | 0.469  | 0.478  |
> > |  | 336 | 0.433  | 0.450  | 0.438  | 0.438  | 0.425  | 0.440  | 0.448  | 0.465  | 0.442  | 0.447  | 0.461  | 0.478  |
> > |  | 720 | 0.481  | 0.485  | 0.453  | 0.466  | 0.448  | 0.469  | 0.464  | 0.477  | 0.456  | 0.470  | 0.495  | 0.497  |
> > |  | Avg | 0.427  | 0.442  | 0.420  | **0.431** | **0.417** | 0.438  | 0.432  | 0.449  | **0.427** | **0.440** | 0.451  | 0.467  |
> > | ETTh2 | 96 | 0.276  | 0.344  | 0.270  | 0.336  | 0.277  | 0.340  | 0.276  | 0.347  | 0.274  | 0.344  | 0.279  | 0.347  |
> > |  | 192 | 0.336  | 0.389  | 0.331  | 0.376  | 0.341  | 0.382  | 0.332  | 0.396  | 0.333  | 0.386  | 0.340  | 0.390  |
> > |  | 336 | 0.340  | 0.398  | 0.325  | 0.387  | 0.331  | 0.387  | 0.347  | 0.410  | 0.338  | 0.403  | 0.342  | 0.406  |
> > |  | 720 | 0.381  | 0.427  | 0.385  | 0.437  | 0.385  | 0.427  | 0.395  | 0.436  | 0.404  | 0.454  | 0.401  | 0.445  |
> > |  | Avg | 0.333  | 0.390  | **0.328** | **0.384** | 0.334  | **0.384** | 0.338  | **0.397** | **0.337** | **0.397** | 0.341  | **0.397** |
> >
> > **Table 3** The comparisons of PatchTST and LTSF-Conv models with look-back window size of {720, 1600} on LTSF benchmarks.
> >
> > See the supplementary materials for the full results.

---

> > > ### Author Response · Authors · 2023-11-20
> > > **Response to Reviewer 1dcB**
> > >
> > > Dear Reviewer,
> > >
> > > Greetings! We would like to express our gratitude for your comments during the review process. We have taken great care to address your question points comprehensively and provide responses.
> > >
> > > **We look forward to your further feedback, and if there are any aspects that you still find unclear or require additional elaboration, we would be more than willing to engage in further discussion**. Your feedback is crucial for us to revise the final manuscript.

---

### Official Review · Reviewer_dk3x · 2023-10-31

**Soundness:** 3 good
**Presentation:** 3 good
**Contribution:** 3 good
**Rating:** 8
**Confidence:** 4

**Summary:**

This paper presents an innovative depthwise convolution model to perform long-term time series forecasting. The key idea is to apply unique filters to each channel to achieve channel independence. The experiment results on public benchmark datasets justified the effectiveness of the proposed method.

**Strengths:**

1. This paper is well written and organized.
2. The proposed convolution based long-term forecasting technique is well-motivated based on a theoretical insight over the periodicity assumption of the time series.
3. Applying RevIN over the one-depthwise convolution operation to deal each channel independently is new. Based on that, a simple yet effective LTSF-Conv models for long term forecasting tasks is developed.
4. The experiment results are comprehensive and quite solid in this paper. State-of-the-art transformer based methods such as PatchTST and MLP-based model TiDE are both compared. The proposed Convolution-based models significant outperform baselines on most cases. In addition, they also consume small GPU memory and exhibit less trainable parameters.

**Weaknesses:**

1. Whether the Conv-LTSF still works for time series that does not exhibit strong periodicity?
2. I wonder whether explicitly considering the channel dependencies can help further improve the forecasting performance.

**Questions:**

Please see the weaknesses

---

> ### Author Response · Authors · 2023-11-18
> **Response to Reviewer dk3x**
>
> We sincerely appreciate your comments and positive ranking. In order to ensure an unbiased evaluation, we rigorously adhered to an identical code structure as that of the other baseline models. This encompassed utilizing the same data preprocessing procedures, including the data provider module. Our contribution was seamlessly integrated by adding our method to the model folder. ALL experimental results can be repeated by utilizing the provided source code. This simple basic unit serves as a robust starting point, showcasing its potential in handling LTSF tasks.
>
> We provide point-wise responses to your concerns below.
>
> Q1 : *Whether the Conv-LTSF still works for time series that does not exhibit strong periodicity?*
> *  Many thanks for your careful reading and suggestion. Following the reviewer’s suggestion, we compared the proposed model (Conv) with the DLinear and strongest Transformer baseline (PatchTST) on the illness benchmark. The illness benchmark describes the ratio of patients seen with illness and the number of patients from 2002 to 2021. It includes weekly data from the Centers for Disease Control and Prevention of the United States (https://gis.cdc.gov/grasp/fluview/fluportaldashboard.html).  Table 4 (in the attached PDF file) provides the corresponding forecasting results. It can be found that Conv outperforms DLinear and PatchTST for all horizons.
>
> | Methods   |Conv|      | PatchTST|      | DLinear|      |
> |:---:|:---:|:---:|:---:|:---:|:---:|:---:|
> | Metric    | MSE|MAE| MSE| MAE  | MSE     | MAE  |
> | 24        | 0.451 | 0.552| 0.637| 0.552| 0.725   | 0.681|
> | 36        | 0.508 | 0.574| 0.765| 0.634| 0.792   | 0.744|
> | 48        | 0.597 | 0.636| 0.756 | 0.692| 0.886   | 0.815|
> | 60        | 0.675 | 0.680| 0.776 | 0.741| 0.960   | 0.859|
> | Avg       | **0.558** | **0.611** | 0.734 | 0.655 | 0.841 | 0.775 |
>
> **Table 4**: The forecasting results on the ILI benchmark under the setting, where sl = 104.
>
> Q2 : *I wonder whether explicitly considering the channel dependencies can help further improve the forecasting performance.*
> * In time series forecasting tasks, we intuitively think that channel-mixing techniques are often a superior approach, which allows the model to effectively capture the interdependencies and interactions between different channels. In channel dependencies, information from various channels is integrated efficiently, facilitating a more comprehensive representation of the complex relationships within the data. However, previous studies [2] have found that channel independence can improve performance compared to channel mixing in LTSF.
> * CI: so-called channel independence; CD: so-called channel dependencie. Following the reviewer’s suggestion, we apply channel independence and channel dependence to the different models respectively. Table 5 (in the attached PDF file) summarizes the results of different channel strategies on the Weather and ETTm1 datasets. **Most of the CI-based models have a higher testing accuracy than the CD-based models.** For MLP-based models, the overall accuracy of the CI-based models was approximately 1\%–8\% higher than the CD-based models.
> * For our models, we have conducted additional ablation experiments about depth-wise CNN (channel independent) and general CNN (channel dependent). The results are in Section C.2, Page 13 of the original manuscript. It can be found that CI-based models also outperform CD-based models.
>
>
> |  |  | |  | Weather  |  |  | |  | ETTm1  |  |  |
> |:---:|:---:|:---:|:---:|:---:|:---:|:---:|:---:|:---:|:---:|:---:|:---:|
> |  |  | 96 | 192 | 336 | 720 | Avg | 96 | 192 | 336 | 720 | Avg |
> | RLinear_CD | MSE | 0.175 | 0.217 | 0.265 | 0.328 | 0.246 | 0.301 | 0.34 | 0.373 | 0.43 | 0.361 |
> |  | MAE | 0.225 | 0.259 | 0.293 | 0.339 | 0.279 | 0.342 | 0.366 | 0.385 | 0.417 | 0.377 |
> | RLinear_CI | MSE | 0.146 | 0.189 | 0.241 | 0.313 | **0.222** | 0.289 | 0.332 | 0.368 | 0.426 | **0.353** |
> |  | MAE | 0.194 | 0.234 | 0.274 | 0.327 | **0.257** | 0.335 | 0.361 | 0.38 | 0.413 | **0.372** |
> | DLinear_CD | MSE | 0.175 | 0.219 | 0.265 | 0.323 | 0.245 | 0.299 | 0.335 | 0.369 | 0.424 | 0.356 |
> |  | MAE | 0.237 | 0.282 | 0.318 | 0.361 | 0.299 | 0.343 | 0.365 | 0.386 | 0.42 | 0.378 |
> | DLinear_CI | MSE | 0.146 | 0.19 | 0.243 | 0.317 | **0.224** | 0.285 | 0.327 | 0.367 | 0.428 | **0.351** |
> |  | MAE | 0.212 | 0.257 | 0.301 | 0.358 | **0.282** | 0.334 | 0.358 | 0.383 | 0.417 | **0.373** |
> | NLinear_CD | MSE | 0.181 | 0.225 | 0.27 | 0.339 | 0.253 | 0.305 | 0.348 | 0.375 | 0.433 | 0.365 |
> |  | MAE | 0.232 | 0.268 | 0.3 | 0.348 | 0.287 | 0.347 | 0.375 | 0.388 | 0.421 | 0.382 |
> | NLinear_CI | MSE | 0.146 | 0.189 | 0.242 | 0.321 | **0.224** | 0.293 | 0.337 | 0.379 | 0.435 | **0.361** |
> |  | MAE | 0.196 | 0.238 | 0.28 | 0.335 | **0.262** | 0.341 | 0.367 | 0.39 | 0.422 | **0.38** |
>
> **Table 5**: Multivariate prediction results on two benchmarks with an input length of 336. CI denotes channel-independence, and CD represents channel-dependence.

---

> > ### Comment · Reviewer_dk3x · 2023-11-19
> >
> > The authors have addressed my concerns.

---

### Official Review · Reviewer_3Fmc · 2023-10-31

**Soundness:** 3 good
**Presentation:** 2 fair
**Contribution:** 2 fair
**Rating:** 6
**Confidence:** 3

**Summary:**

**Summary:**

The paper aims to address the challenges faced by Transformer-based models in long-term time series forecasting (LTSF) tasks, mainly when dealing with long sequence lengths. The authors propose a new model, LTSF-Conv, which utilizes depthwise convolution models to enhance forecasting performance while significantly reducing computational costs.

**Strengths:**

From the computational efficiency (memory usage/flops) perspective, this paper reports very promising results on several datasets.

**Weaknesses:**

From the accuracy perspective, two recent CNN baseline models, TimesNet and MICN [1] are missing in Table 1 and Table 2. Moreover, the hyperparameter-searching is used, which makes the comparison a little bit unfair. For example, in TimesNet and MICN, the model configurations and lookback window remain the same for most of the experiments, and in PatchTST, only two configurations are considered. Based on the current results, it is hard for me to tell whether the performance gain is from the better model configuration or the proposed structure.

Moreover, based on my understanding of Section 4.1, the main takeaway message would be there are two useful structures, depth-wise 1Dconv, and/or trend/seasonality decomposition. A similar idea (i.e., 1Dconv + decomposition) is also mentioned in MICN (e.g.,  Figure 1 in [1]). One more interesting thing here is the usage of depth-wise CNN instead. As shown in Table 4, deep-wise gives significant performance improvements. I would expect a more in-depth analysis of why it reaches better results than vanilla CNN. Based on the current presentation, it is a little bit difficult for me to understand what inductive bias can only be utilized by depth-wise CNN but not general CNN.

The theoretical analysis is also kind of weak. Theorem 1 and Corollary 1 consider simple autoregressive state-space structure and MLP/RNN models can also have the same prediction power. MLP can be viewed as a CNN with kernel size equal to the sequence length. RNN is commonly used to model state-space structures. Theorem 2 considers the sequence with both trend and seasonality. From my understanding, the MLP/RNN may also reach a similar performance guarantee.

After reviewing the sample codes in the supplementary material. I also have some concerns about the numerical results reported in the paper. When dealing with test samples, the data_provider function sets the drop_last = True and shuffle_flag = False. The consequence would result in the last several test samples being ignored. Those samples are usually the hardest to predict since they are far away from the training set. Moreover, it seems the main results in Table1 and Table2 are only run with one fixed random seed 1024. The random control experiment is only reported in Figure 5 in the Appendix.


**Questions and Suggestions:**

1. As the title used the word *minimalism*, I would conjecture the main advantage of using simple depth-wise CNN would be its robustness. The time series forecating usually contains a lot of time-varying noise especially when using longer inputs. The usage of a simpler model would have less risk of overfitting that noise but a potential drawback would be more modeling bias may be introduced due to limited representation power. Therefore, I would expect the analysis from the theoretical part to consider the high noise system, such as $x(t)  = x(t-p) + \epsilon\_t$ where $\epsilon\_t$  could be on the same order of $x(t)$, and analyze the generalization ability of depth-wise CNN to show it will have better variance bias trade-off.

2. Please add TimesNet and MICN as benchmarks in Table 1 and Table 2.

3. Please fix the dataloader issue in the test part and rerun the relevant experiments. It would be better to also report the random control results in Table 1 and Table 2.

4. Please provide the detailed experimental configurations for each setting in Table 1 and Table 2 to help the reviewer verify those results.

5. Could the author elaborate more on the seq_last in ConvNet.py file? It seems not to be discussed in Section 4. Moreover, since Revnorm has been used, the sequence would already be centered, why do we still need to subtract the sequence mean?


**Conclusion:**

While the paper explores an intriguing concept that simpler models might suffice for certain datasets, the current depth of analysis and the reliability of numerical results do not yet support a strong case for acceptance at a top-tier machine learning conference like ICLR. Despite this, the reviewer is willing to reconsider the decision after the authors' rebuttal.




**Reference**

[1] Wang, Huiqiang, Jian Peng, Feihu Huang, Jince Wang, Junhui Chen, and Yifei Xiao. "Micn: Multi-scale local and global context modeling for long-term series forecasting." In The Eleventh International Conference on Learning Representations. 2022.

**Strengths:**

Please refer to the Strengths section in Summary.

**Weaknesses:**

Please refer to the Weaknesses section in Summary.

**Questions:**

Please refer to the Questions and Suggestions section in Summary.

---

> ### Author Response · Authors · 2023-11-18
> **Response to Reviewer 3Fmc**
>
> We appreciate that you found some strengths. Before responding point-by-point, we would like to provide some insights into the contribution of our work. In its initial publication, the DLinear model [1] faced widespread controversy due to its one-layer linear model structure. Despite this initial skepticism, the model ultimately achieved success and exerted a significant influence within the LTSF community. Based on this research, lots of variant models based on MLPs have emerged. However, our model structure is as simple as DLinear, while extensive experiments have consistently demonstrated that depth-wise convolutional units possess inherent advantages over Linear units of DLinear in LTSF. **We firmly believe that LTSF-Conv models serve as straightforward yet competitive basic units, exhibiting great potential for further expansion of complex network structures.**
>
> Q1 : *From the accuracy perspective, two recent CNN baseline models, TimesNet and MICN [1] are missing in Table 1 and Table 2. Moreover, the hyperparameter-searching is used, which makes the comparison a little bit unfair. For example, in TimesNet and MICN, the model configurations and lookback window remain the same for most of the experiments, and in PatchTST, only two configurations are considered. Based on the current results, it is hard for me to tell whether the performance gain is from the better model configuration or the proposed structure.*
>
> * Thank you for pointing out this. We have confidence in the fairness of our previous and future comparisons. First, **the experimental results not only include Conv-Best and DConv-Best but also the default look-back window length is set at 512, such as Conv and Dconv in Table 1 and Table 2 of the original manuscript.** This is using the same lookback window length as PatchTST (sl=512). Then, Transformer-based baselines have not benefited from a longer look-back window. The reason why most transformer models do not increase the lookback window length is because the performance will become worse. More analysis of the look-back windows is described in Section C.1, Page 12. in the original Appendix. Reference [1] has also proved it.
> * PatchTST can extend the lookback window to 512, but does not benefit from longer windows. It's important to note that this extension comes at the cost of a substantial increase in the model’s computational resource requirements. On our server resources, **PatchTST ran out of GPU memory for a look-back window size greater than 720.** Reference [3] also conducted similar experiments, encountering the same issue of OOM. This is also a significant limitation of the PatchTST model.
> * Following the reviewer’s suggestion, we switched to a higher-performance server to validate PatchTST with lookback-window size $\in$ \{720, 1600\}. The results are in Table 3 of the attached supplementary material. **It can be found that not only the overall performance has not improved, but the training cost is almost unacceptable.**
> * Finally, We have added MICN and TimesNet baseline models. We adopt their official codes and only change the length of input sequences. **The results are summarized in Table 2 of the attached PDF file.**  In Table 2, we conducted an experiment within a broader window size range of \{96, 336, 512, 720, 1600\}. We consistently chose the most optimal results, thus establishing robust baselines. The results demonstrate that LTSF-Conv models consistently surpass all MICN and TimesNet on seven LTSF benchmarks.
>
> Q2 : *I would expect a more in-depth analysis of why it reaches better results than vanilla CNN.*
>
> * Thank you for pointing out this. In LTSF tasks, channel independence can enhance prediction performance compared to channel mixing. Previous research  [2] has also found it. For MLP-based variants, achieving channel-independent techniques typically involves multiple independent MLP sub-models, each responsible for handling a specific channel or feature within the time series. It comes at the cost of a substantial increase in the model’s computational resource requirements. As a comparison, our model utilizes group-wise convolution to cleverly achieve channel independence, while concurrently reducing model complexity. This is because depthwise convolution exhibits higher efficiency than standard convolution. In the context of group-wise convolution, a critical consideration lies in aligning the number of channels with the variable dimension and the number of filters, as defined in the initial setup. However, standard CNN uses the idea of channel mixing, which suffers from noise interference among the channels and reduces performance. We have conducted additional ablation experiments about depth-wise CNN and general CNN. The results are in Section C.2, Page 13 of the original manuscript.

---

> > ### Author Response · Authors · 2023-11-18
> > **Response to Reviewer 3Fmc**
> >
> > Q3 : *Please provide the detailed experimental configurations for each setting in Table 1 and Table 2 to help the reviewer verify those results.*
> > * Thank you for your comments. We add Table 1 in the attached supplementary material, which provides detailed experimental configurations of Conv with sl=512 on all datasets. In the original paper, patchTST also adopted this step size of 512.
> >
> > Q4 : *I also have some concerns about the numerical results reported in the paper. When dealing with test samples, the data\_provider function sets the drop\_last = True and shuffle\_flag = False. The consequence would result in the last several test samples being ignored. Those samples are usually the hardest to predict since they are far away from the training set. Moreover, it seems the main results in Table1 and Table2 are only run with one fixed random seed 1024. The random control experiment is only reported in Figure 5 in the Appendix. Why do we still need to subtract the sequence mean?*
> > * Thank you for pointing out this. We have confidence in the fairness of our previous comparison. To ensure an objective evaluation, we strictly adhered to the same code structure as the other baseline models, which includes using the same ‘data\_factory’ and ‘data\_loader’ files, and we just added the file of the proposed method in the model folder.  Test\{'shuffle\_flag:' False, 'drop\_last: True' \}; Pred\{'shuffle\_flag:' False, 'drop\_last: False' \}; Else\{'shuffle\_flag:' True, 'drop\_last: True' \} The training set, test set, and validation set for all datasets remain consistent with the previous experiments. Reviewers can check the PatchTST and related models' source code to thoroughly verify the fairness of the experiment.
> >
> > * Moreover, Table 1 and Table 2 of the original manuscript present the average results obtained from three experiments using different random seeds (Conv-Best, DConv-Best). It can be found in Section B, page 12 of the original manuscript.
> >
> > * Figure 5 shows robust experimental results in the original manuscript. To obtain a better visualization quality, each experimental configuration was repeated with five random seeds. Since Revnorm has been applied, the value of seq\_last is 0, and subtracting it has no impact on the input variable $X$. Of course, it will not affect the prediction results. It seems there was an upload error, and the final released code will be updated.

---

> > > ### Author Response · Authors · 2023-11-20
> > > **Response to Reviewer 3Fmc**
> > >
> > > We add detailed experimental configurations for univariate results in Table 2 of the original manuscript. Reviewers can further verify the experimental results.
> > >
> > > | Dataset| Seq_Len| Pre_Len| Batch_Size| Learning_Rate| Kernel_Size| Individual| Mse| Mae|
> > > |:---:|:---:|:---:|:---:|:---:|:---:|:---:|:---:|:---:|
> > > | ETTh1 | 512 | 96 | 16 | 0.005 | 24 | 0 | 0.053 | 0.177 |
> > > |  | 512 | 192 | 16 | 0.005 | 55 | 0 | 0.064  | 0.197 |
> > > |  | 512 | 336 | 64 | 0.005 | 55 | 0 | 0.075  | 0.217 |
> > > |  | 512 | 720 | 128 | 0.005 | 78 | 0 | 0.082  | 0.227  |
> > > | ETTh2 | 512 | 96 | 128 | 0.005 | 24 | 0 | 0.134  | 0.284 |
> > > |  | 512 | 192 | 16 | 0.005 | 35 | 0 | 0.172  | 0.328 |
> > > |  | 512 | 336 | 128 | 0.005 | 24 | 0 | 0.179  | 0.342 |
> > > |  | 512 | 720 | 128 | 0.005 | 24 | 0 | 0.219  | 0.376  |
> > > | ETTm1 | 512 | 96 | 64 | 0.005 | 55 | 0 | 0.026  | 0.122  |
> > > |  | 512 | 192 | 64 | 0.005 | 24 | 0 | 0.039  | 0.150  |
> > > |  | 512 | 336 | 16 | 0.005 | 35 | 0 | 0.052  | 0.172  |
> > > |  | 512 | 720 | 16 | 0.005 | 35 | 0 | 0.071  | 0.203  |
> > > | ETTm2 | 512 | 96 | 16 | 0.005 | 55 | 0 | 0.062  | 0.182  |
> > > |  | 512 | 192 | 16 | 0.005 | 24 | 0 | 0.090  | 0.225  |
> > > |  | 512 | 336 | 64 | 0.005 | 24 | 0 | 0.118  | 0.261  |
> > > |  | 512 | 720 | 16 | 0.005 | 24 | 0 | 0.172  | 0.320  |
> > >
> > > **Table 6**: The hyperparameters of Conv modele with look-back window size 512 on ETT datasets. Note that the default training epochs are set to 100.
> > >
> > >
> > > | Methods|  | Conv|  | DConv|  | TimesNet*|  | MICN-regre*|  |
> > > |:---:|:---:|:---:|:---:|:---:|:---:|:---:|:---:|:---:|:---:|
> > > | |  | MSE | MAE | MSE | MAE | MSE | MAE | MSE | MAE |
> > > | Weather | 96 | 0.140  | 0.188  | 0.166  | 0.220  | 0.159  | 0.215  | 0.161  | 0.229  |
> > > |  | 192 | 0.182  | 0.230  | 0.209  | 0.259  | 0.219  | 0.261  | 0.220  | 0.281  |
> > > |  | 336 | 0.237  | 0.271  | 0.253  | 0.293  | 0.274  | 0.306  | 0.257  | 0.316  |
> > > |  | 720 | 0.294  | 0.324  | 0.306  | 0.335  | 0.347  | 0.356  | 0.311  | 0.356  |
> > > |  | Avg | **0.213** | **0.253** | 0.234  | 0.277  | 0.250  | 0.285  | 0.237  | 0.296  |
> > > | Electricity | 96 | 0.129  | 0.225  | 0.130  | 0.225  | 0.168  | 0.272  | 0.155  | 0.265  |
> > > |  | 192 | 0.143  | 0.238  | 0.144  | 0.238  | 0.184  | 0.289  | 0.177  | 0.285  |
> > > |  | 336 | 0.159  | 0.255  | 0.160  | 0.256  | 0.196  | 0.299  | 0.180  | 0.292  |
> > > |  | 720 | 0.195  | 0.286  | 0.199  | 0.288  | 0.220  | 0.320  | 0.207  | 0.316  |
> > > |  | Avg | **0.157** | **0.251** | 0.158  | 0.252  | 0.192  | 0.295  | 0.180  | 0.290  |
> > > | ETTm2 | 96 | 0.161  | 0.249  | 0.161  | 0.253  | 0.187  | 0.267  | 0.176  | 0.275  |
> > > |  | 192 | 0.217  | 0.287  | 0.213  | 0.292  | 0.249  | 0.309  | 0.254  | 0.334  |
> > > |  | 336 | 0.257  | 0.329  | 0.258  | 0.325  | 0.295  | 0.349  | 0.288  | 0.351  |
> > > |  | 720 | 0.325  | 0.379  | 0.325  | 0.369  | 0.408  | 0.403  | 0.417  | 0.440  |
> > > |  | Avg | 0.240  | 0.311  | **0.239** | **0.310** | 0.285  | 0.332  | 0.284  | 0.350  |
> > > | ETTm1 | 96 | 0.287  | 0.334  | 0.300  | 0.342  | 0.335  | 0.376  | 0.311  | 0.364  |
> > > |  | 192 | 0.328  | 0.358  | 0.335  | 0.363  | 0.374  | 0.387  | 0.356  | 0.388  |
> > > |  | 336 | 0.356  | 0.384  | 0.356  | 0.384  | 0.410  | 0.411  | 0.407  | 0.422  |
> > > |  | 720 | 0.394  | 0.408  | 0.393  | 0.405  | 0.478  | 0.450  | 0.464  | 0.462  |
> > > |  | Avg | **0.341** | **0.371** | 0.346  | 0.374  | 0.399  | 0.406  | 0.408  | 0.425  |
> > > | ETTh1 | 96 | 0.365  | 0.393  | 0.366  | 0.382  | 0.384  | 0.402  | 0.389  | 0.424  |
> > > |  | 192 | 0.403  | 0.418  | 0.400  | 0.412  | 0.436  | 0.429  | 0.474  | 0.487  |
> > > |  | 336 | 0.424  | 0.428  | 0.421  | 0.422  | 0.491  | 0.469  | 0.516  | 0.524  |
> > > |  | 720 | 0.450  | 0.460  | 0.429  | 0.446  | 0.521  | 0.500  | 0.743  | 0.664  |
> > > |  | Avg | 0.411  | 0.425  | **0.404** | **0.416** | 0.458  | 0.450  | 0.531  | 0.525  |
> > > | ETTh2 | 96 | 0.268  | 0.339  | 0.269  | 0.334  | 0.340  | 0.374  | 0.299  | 0.364  |
> > > |  | 192 | 0.327  | 0.382  | 0.326  | 0.375  | 0.402  | 0.414  | 0.441  | 0.454  |
> > > |  | 336 | 0.329  | 0.390  | 0.321  | 0.386  | 0.390  | 0.437  | 0.654  | 0.567  |
> > > |  | 720 | 0.380  | 0.424  | 0.382  | 0.428  | 0.462  | 0.468  | 0.956  | 0.716  |
> > > |  | Avg | 0.326  | 0.384  | **0.325** | **0.381** | 0.399  | 0.423  | 0.588  | 0.525  |
> > > | Traffic | 96 | 0.383  | 0.271  | 0.378  | 0.264  | 0.593  | 0.321  | 0.473  | 0.306  |
> > > |  | 192 | 0.397  | 0.275  | 0.390  | 0.269  | 0.615  | 0.331  | 0.475  | 0.298  |
> > > |  | 336 | 0.411  | 0.282  | 0.404  | 0.275  | 0.629  | 0.336  | 0.493  | 0.307  |
> > > |  | 720 | 0.450  | 0.302  | 0.442  | 0.294  | 0.640  | 0.350  | 0.531  | 0.325  |
> > > |  | Avg | 0.410  | 0.283  | **0.404** | **0.276** | 0.619  | 0.335  | 0.493  | 0.309  |
> > >
> > > **Table 2**: Multivariate prediction results. * denotes re-implementation after increasing the input length.
> > >
> > > **If there are any areas where our explanations are not clear or if any other questions, please let us know and we expect more discussions.** We want to ensure that you have all the information you need to make a final decision. Your understanding and satisfaction are crucial for us to revise the final Manuscript. Thanks

---

### Author Response · Authors · 2023-11-18

Dear Reviewers, ACs, and PCs,

With this letter, we would like to express our deep appreciation for your tremendous efforts in reviewing our manuscript entitled "The Power of Minimalism in Long Sequence Time-series Forecasting" at the ICLR conference. Those comments are constructive and helpful for improving the paper. Due to the additional experiments involving multiple models on seven public datasets, our submission for the rebuttal is slightly late. **We are confident in the fairness of our previous comparisons. Table 1 provides detailed experimental configurations, and reviewers can verify experiment results**. We present the point-to-point responses to the reviewers' comments below. We hope that our responses have addressed your concerns, and we look forward to engaging in further discussions with you. If you have any additional questions or comments, please feel free to share them before the author-reviewer discussion period concludes. If our responses have satisfactorily resolved your concerns, we kindly request that you consider revising the rating of our work. Thank you once again for your valuable time and efforts.

We add some common references in this section.

[1] Zeng, A., Chen, M., Zhang, L., \& Xu, Q. (2023, June). Are transformers effective for time series forecasting? In Proceedings of the AAAI conference on artificial intelligence (Vol. 37, No. 9, pp. 11121-11128).

[2] Nie, Y., Nguyen, N. H., Sinthong, P., \& Kalagnanam, J. (2023). A Time Series is Worth 64 Words: Long-term Forecasting with Transformers. In The Eleventh International Conference on Learning Representations.

[3] Das, A., Kong, W., Leach, A., Sen, R., & Yu, R. (2023). Long-term Forecasting with TiDE: Time-series Dense Encoder. arXiv preprint arXiv:2304.08424.

| Datasets | Seq_Len | Pre_Len | Batch_Size | Learning_Rate | Kernel_Size | Individual | MSE | MAE |
|:---:|:---:|:---:|:---:|:---:|:---:|:---:|:---:|:---:|
| ETTh1 | 512 | 96 | 16 | 0.005 | 55 | 0 | 0.365  | 0.393  |
|  | 512 | 192 | 16 | 0.005 | 55 | 0 | 0.401  | 0.416  |
|  | 512 | 336 | 128 | 0.005 | 78 | 1 | 0.419  | 0.437  |
|  | 512 | 720 | 16 | 0.005 | 55 | 0 | 0.464  | 0.472  |
| ETTh2 | 512 | 96 | 128 | 0.005 | 55 | 0 | 0.269  | 0.339  |
|  | 512 | 192 | 128 | 0.005 | 78 | 0 | 0.329  | 0.383  |
|  | 512 | 336 | 128 | 0.0005 | 78 | 0 | 0.335  | 0.394  |
|  | 512 | 720 | 128 | 0.0001 | 78 | 0 | 0.379  | 0.424  |
| ETTm1 | 512 | 96 | 16 | 0.005 | 78 | 1 | 0.292  | 0.338  |
|  | 512 | 192 | 16 | 0.005 | 35 | 0 | 0.332  | 0.361  |
|  | 512 | 336 | 16 | 0.005 | 35 | 0 | 0.364  | 0.380  |
|  | 512 | 720 | 16 | 0.005 | 35 | 0 | 0.418  | 0.411  |
| ETTm2 | 512 | 96 | 16 | 0.005 | 24 | 1 | 0.161  | 0.249  |
|  | 512 | 192 | 16 | 0.005 | 55 | 1 | 0.216  | 0.288  |
|  | 512 | 336 | 16 | 0.005 | 35 | 0 | 0.271  | 0.327  |
|  | 512 | 720 | 64 | 0.005 | 24 | 0 | 0.361  | 0.387  |
| Weather | 512 | 96 | 16 | 0.005 | 55 | 0 | 0.140  | 0.188  |
|  | 512 | 192 | 16 | 0.005 | 78 | 1 | 0.183  | 0.230  |
|  | 512 | 336 | 16 | 0.005 | 24 | 1 | 0.234  | 0.271  |
|  | 512 | 720 | 128 | 0.005 | 78 | 0 | 0.306  | 0.325  |
| Electricity | 512 | 96 | 16 | 0.005 | 55 | 0 | 0.132  | 0.227  |
|  | 512 | 192 | 128 | 0.005 | 55 | 1 | 0.145  | 0.241  |
|  | 512 | 336 | 64 | 0.005 | 55 | 0 | 0.161  | 0.257  |
|  | 512 | 720 | 16 | 0.005 | 55 | 0 | 0.201  | 0.289  |
| Traffic | 512 | 96 | 16 | 0.005 | 35 | 0 | 0.396  | 0.275  |
|  | 512 | 192 | 16 | 0.005 | 35 | 0 | 0.407  | 0.279  |
|  | 512 | 336 | 16 | 0.005 | 24 | 0 | 0.417  | 0.285  |
|  | 512 | 720 | 16 | 0.005 | 24 | 0 | 0.453  | 0.304  |

**Table 1**: The hyperparameters of Conv modele with look-back window size 512. Note that the default training epochs are set to 100.

---

### Comment · Area_Chair_LQqV · 2023-11-23
**From AC at the end of rebuttal: Reviewer response required**

Dear Reviewers,

Thanks for your time and commitment to the ICLR 2024 review process.

As we approach the conclusion of the author-reviewer discussion period (Wednesday, Nov 22nd, AOE), I kindly urge those who haven't engaged with the authors' dedicated rebuttal to please take a moment to review their response and share your feedback, regardless of whether it alters your opinion of the paper.

Your feedback is essential to a thorough assessment of the submission.

Best regards,

AC

---

### Meta-Review · Area_Chair_LQqV · 2023-12-11

**Metareview:**

This paper presents a simple convolution-based solution to time series forecasting, which is coined as "minimalism" in the title. The claim of "minimalism" is subjective as there is no possible explanation of how simple is "minimalism." Authors performed an extensive rebuttal, which was appreciated by the reviewers with post-rebuttal scores raised. In general, most of the concerns were addressed, while there was still a score of "3". The AC's take on this paper is that the current paper provides a poor presentation, as can be witnessed by scanning throughout the paper---it is subpar to normal ICLR papers, in that the content in page 3 is loosely placed, the numbers in the tables are densely listed, the figures are produced with low readability. Besides, the technical contribution is not overwhelming as argued by the authors---these days, there are many attempts in achieving "simple" models for time series forecasting, however, they were later shown to be inferior to more sophisticated method such as PatchTST, if all models were to be evaluated fairly and hyperparameter-tuned carefully. Expecting solid evaluation is usually difficult in this line of methods, because complex tasks that are more relevant are bypassed otherwise the performance cannot be convincing. While this work can be used as a simple baseline, this field does not need yet another one. In summary, this is a decent work but is below the bar of this conference, and it will not draw significant impact from the community.

**Justification For Why Not Higher Score:**

A decent work, but not good enough. Presentation quality is clearly subpar. The technical solution makes sense but does not make a worthwhile contribution. Evaluation is not convincing enough, in the sense of using more complex tasks and better tuned protocols.

**Justification For Why Not Lower Score:**

N/A

---

### Decision · Program_Chairs · 2024-01-16

Reject